# Determining Homogenization Parameters and Predicting 5182-Sc-Zr Alloy Properties by Artificial Neural Networks

**DOI:** 10.3390/ma16155315

**Published:** 2023-07-28

**Authors:** Jingxiao Li, Dongfang Du, Xiaofang Yang, Youcai Qiu, Shihua Xiang

**Affiliations:** 1Department of Materials Engineering, Sichuan Engineering Technical College, Deyang 618000, China; lijingxiao@scetc.edu.cn (J.L.); ddf@scetc.edu.cn (D.D.); 2International Joint Laboratory for Light Alloys (Ministry of Education), College of Materials Science and Engineering, Chongqing University, Chongqing 400044, China; qiuyoucai@cqu.edu.cn (Y.Q.); xiangsh@cqu.edu.cn (S.X.)

**Keywords:** 5182-Sc-Zr alloy, artificial neural network, homogenization treatment, recrystallization treatment, parameters selection

## Abstract

Artificial neural networks (ANNs) were established for the homogenization and recrystallization heat treatment processes of 5182-Sc-Zr alloy. Microhardness and conductivity testing were utilized to determine the precipitation state of Al_3_(Sc_x_Zr_1−x_) dispersoids during the homogenization treatment, while electron backscatter diffraction (EBSD) and transmission electron microscopy (TEM) were used to observe the microstructure evolution of the alloy. Tensile experiments were performed to test the mechanical properties of the alloy after recrystallization annealing. The two-stage homogenization parameters were determined by studying the changes in microhardness and electrical conductivity of 5182-Sc-Zr alloy after homogenization with the assistance of artificial neural networks: the first-stage homogenization at 275 °C for 20 h and the second-stage homogenization at 440 °C for 12 h. The dispersoids had entirely precipitated after homogenization, and the alloy segregation had improved. A high-accuracy prediction model, incorporating multiple influencing factors through artificial neural networks, was successfully established to predict the mechanical properties of the 5182-Sc-Zr alloy after annealing. Based on the atomic plane spacing in HRTEM, it was determined that the Al_3_(Sc_x_Zr_1−x_) dispersoids and the Al matrix maintained a good coherence relationship after annealing at 400 °C.

## 1. Introduction

In previous research, the primary reason for the impact of Sc and Zr microalloying on the microstructure and properties of aluminum alloys is the formation of Al_3_(Sc_x_Zr_1−x_) dispersoids [1,2,3,4]. This nanoscale dispersoid is precipitated during the homogenization treatment of Al-Sc-Zr alloy ingots due to the supersaturated solid solution decomposition [5]. At different homogenization methods, the precipitation state of the dispersoids varies, which can affect the subsequent properties of the Al alloy [6]. During the subsequent deformation process, the Al_3_(Sc_x_Zr_1−x_) dispersoids can effectively impede dislocations, sub-grain boundaries, and grain boundaries’ motion [7]. During the recrystallization annealing treatment, the pegging effect of the dispersoids on dislocations can enhance the recrystallization resistance of the alloy, and the synergistic effect of the incompletely recrystallized microstructure and the dispersoids can affect the mechanical properties of the final product [8]. During heat treatment and processing, the state of Al_3_(Sc_x_Zr_1−x_) dispersoids changed, which consequently resulted in alterations to the microstructure and properties of the alloy [9]. When the Al_3_(Sc_x_Zr_1−x_) dispersoids precipitate, the parameters of the homogenization treatment need to be controlled to prevent them from thermal coarsening, and the effect of the coarsening on the strengthening effect needs to be considered after the subsequent recrystallization annealing. It can be seen that the Al_3_(Sc_x_Zr_1−x_) dispersoids are affected by the homogenization treatment and the final annealing treatment, i.e., the final mechanical properties are affected by both.

Homogenization treatment and final annealing treatment bring about multi-threaded and non-linear effects on the microstructure and performance of the alloy. Regarding the non-linear relationship of multi-threading, the limitations of mathematical models may result in a decrease in prediction accuracy [10]. Therefore, some researchers have begun to introduce machine learning methods into material research to establish models [11,12,13,14]. The wide variety of materials and the complexity of influencing factors faced in materials research can be solved by neural networks brought about by big data-driven science [15]. Neural networks are adept at discovering connections among massive amounts of data, and introducing this approach is helpful for understanding and discovering nonlinear relationships between experimental parameters, material microstructure, and properties. Currently, ANNs have been successfully applied in various fields of material research. In Jiang’s study, the utilization of the backpropagation neural network in conjunction with the finite element method yielded outcomes for the accurate forecasting of 2A12 alloy grain size subsequent to extrusion [16]. In the research conducted by Kuppusamy on eco-friendly engineered geopolymer composites, the employment of an artificial neural network with cross-validation methodology aided in the formulation of the material design, which led to the realization of the intended compressive and tensile strength properties in the final product [17]. In the investigations pertaining to the thermal deformation properties of metals, researchers developed constitutive models, strain-compensated constitutive models, and artificial neural networks for the purpose of flow stress prediction. Through the assessment of the models’ predictive capabilities using residual analysis, correlation coefficient (R), and average absolute relative error (AARE), it was determined that the artificial neural network exhibited superior accuracy compared to other models [18,19]. Artificial neural networks have been applied in the field of materials science to some extent. However, compared to the scale of data in this field, the current research is still relatively small, and the models cannot yet be used for production guidance. Therefore, it is necessary to enrich and improve research in this aspect.

In this paper, the effect of Al_3_(Sc_x_Zr_1−x_) dispersoids precipitation on the properties of 5182-Sc-Zr alloy during double-stage homogenization annealing was studied. Regarding this research, artificial neural networks can be used to assist in selecting parameters for homogenization treatment. An artificial neural network was used to predict the microhardness and electrical conductivity of 5182-Sc-Zr alloy under different homogenization conditions, and the predicted data were used to select an appropriate homogenization process. The influence of Al_3_(Sc_x_Zr_1−x_) dispersoids on the recrystallized microstructure and room temperature mechanical properties of the 5182-Sc-Zr alloy was then investigated. The majority of studies utilizing neural networks to establish models have predominantly focused on single production or experimental processes, thereby being limited to capturing the impact of processing conditions or heat treatment conditions on alloy performance within a singular process [16,17,18,19]. In this study, the homogenization treatment parameters, recrystallization annealing treatment parameters, and corresponding tensile properties of the 5182-Sc-Zr alloy can be used to build a correlated artificial neural network with high generalization ability, and this network, covering multiple influences, is used to reflect the response of mechanical properties to the synergistic effect of multi-threaded influences.

## 2. Materials and Methods

### 2.1. Materials

The average chemical composition of alloys, which was measured by optical emission spectroscopy, is shown in Table 1. In this study, three kinds of alloys were investigated: one was 5182 alloy containing 0.05 wt.% Sc and 0.1 wt.% Zr (referred to as 5182-0.05Sc-0.1Zr alloy hereafter), one was 5182 alloy containing 0.1 wt.% Sc and 0.1 wt.% Zr (referred to as 5182-0.1Sc-0.1Zr alloy hereafter), and AA5182 was used as a reference alloy.

### 2.2. Homogenization Treatment

As-cast specimens with a diameter of 5 mm and a thickness of 1.5 mm were subjected to differential scanning calorimetry analysis in a Mettle-Toledo TGA/DSC (Mettler Toledo Technology (China) Co., Ltd., Shanghai, China) type thermal analyzer, heated from room temperature at a rate of 20 °C/min with an argon atmosphere. The theoretical over-burning temperature was determined from the obtained heat absorption and exothermic curves, which led to the determination of an upper limit of 560 °C for the homogenization temperature used in this study. The homogenization temperature range was developed with reference to previous studies [20,21,22,23]. To explore the optimal first stage and second stage homogenization temperatures, we investigated the variation of alloy properties from 200 °C to 350 °C and 400 °C to 480 °C, respectively.

The homogenization treatment was conducted using an air furnace, and a rate of 30 °C/h was used to warm up the samples. After the furnace temperature reached the predetermined holding temperature, the samples were taken out at certain time intervals and water quenched immediately. The surface of the water-quenched sample was ground to 4000# sandpaper, after which electrical conductivity and microhardness were tested. The electrical conductivity testing was performed by SIGMASCOPE SMP10 (Nantong Fischer Instrumentation Ltd., Shanghai, China). The microhardness was obtained by Shimadzu HMV-G (Shimadzu Instrument (Suzhou) Co., Ltd., Suzhou, China) with a load of 0.2 kgf. The heat treatment flow is shown in Figure 1.

### 2.3. Recrystallization Annealing Treatment

The as-homogenized 5182-Sc-Zr alloys were annealed after hot rolling and cold rolling. Among them, the heating temperature of hot rolling is 510 °C, held in the furnace for 40 min before rolling. After multi-passes of cold rolling, sheets with a cold rolling reduction of 70% and a thickness of 3 mm are obtained. In previous studies on the recrystallization of 5182 alloy, the annealing temperatures were in the range of 200 °C to 540 °C [24]. In this study, four temperatures in this range were selected for testing. The cold-rolled sheets were annealed at 250 °C, 300 °C, 400 °C, and 500 °C for 2 h, followed by water quenching, respectively.

### 2.4. Tensile Property Test

The cold-rolled and annealed sheets were tested for tensile testing along the rolling direction at a strain rate of 1 mm/min using Shimadzu AGX (50 KN). Strain is recorded using a calibrated linear variable differential transformer attached to an optical automatic extensometer. Stress–strain data are collected via a computerized data acquisition system.

### 2.5. Microstructural Analysis

EBSD and EDS data were collected on a Zeiss focused ion beam scanning electron microscopes (FIB/SEM) to calibrate the longitudinal sections of annealed sheets at 20 kV and 15 kV, respectively. Al_3_(Sc_x_Zr_1−x_) dispersoids were observed using a FEI Talos F200s (Thermo Fisher Scientific Inc., Shanghai, China) transmission electron microscope with an operating voltage of 200 kV.

### 2.6. Establishment of Artificial Neural Networks

The artificial neural networks established in this study were implemented with the assistance of MATLAB. The input layer data and output layer data should be determined before building the neural network, and then the number of hidden layers and the number of hidden layer neurons should be set in the process of building the neural network. To improve the speed and accuracy of network training, all data are normalized before building the network. For the output data, an inverse normalization process is required to compare it with the original data and obtain the model error.

For the homogenization input layer, the two-stage method must be taken into account, which implies that the parameters for the second stage are given a value of “0” for samples that have only undergone the first-stage homogenization. The input data include the homogenization temperature, holding duration, and contents of Sc and Zr. Microhardness and electrical conductivity are set as output data. The data are adjusted before training by setting the temperature T (°C) to 1/T (1/K), the time t (h) to ln(t) (h), and the microhardness H (HV) to H/H_max_.

For the room temperature properties model, the temperature and duration of the two-stage homogenization, the strain of the cold deformation prior to annealing, the contents of Sc and Zr, and the recrystallization annealing temperature are set as input data. The output data are the corresponding stresses. After normalization, the stress is transformed from σ (MPa) to σ/σ_max_.

## 3. Results and Discussion

### 3.1. Selection of Homogenization Parameters with ANN Assistance

The microhardness of the 5182-Sc-Zr alloy did not vary much throughout the 200 °C homogenization treatment, as shown in Figure 2. When the temperature was elevated to 250 °C, microhardness of the 5182-Sc-Zr alloy started to increase at 16 h, but the fluctuation of microhardness persisted until the homogenization time reached 40 h. The microhardness of the 5182-Sc-Zr alloys increased after 1 h of holding at 300 °C and 350 °C, then stabilized after about 16 h.

Figure 3 displays the electrical conductivity variation during the first stage homogenization. At 200 °C, the 5182-Sc-Zr alloy’s conductivity barely changed. From 250 °C to 350 °C, the conductivity increases steadily, reaching a peak of 4.8% at 350 °C.

The primary objective of the first-stage homogenization treatment is to promote the dissolution of the supersaturated solid solution containing Sc into the Al_3_Sc dispersoids. The microhardness of the 5182-Sc-Zr alloy is increased by the strengthening effect of Al_3_Sc [25]. Meanwhile, as Sc atoms precipitate, fewer solid solution atoms remain in the Al matrix, increasing the conductivity of the 5182-Sc-Zr alloy. The precipitation state of the Al_3_Sc dispersoids can be inferred by determining changes in microhardness and conductivity.

Based on changes in microhardness and conductivity, the Al_3_Sc dispersoids did not precipitate in the 5182-Sc-Zr alloy at 200 °C; hence, this temperature is not suitable for the first-stage homogenization annealing treatment. Additionally, the microhardness of the alloy 5182-0.1Sc-0.1Zr failed to stabilize after a 40-h treatment, making the first-stage homogenization annealing treatment inappropriate at 250 °C. It has been discovered that the first-stage homogenization annealing treatment temperature should be greater than 250 °C, and further research should be carried out at temperatures ranging from 250 °C to 350 °C. Then, an artificial neural network was created to choose the homogenization parameter.

Before building an artificial neural network based on the first-stage homogenization treatment, the data set needs to be normalized. According to the aforementioned study, the parameters impacting the microhardness comprise the addition of Sc and Zr, as well as the treatment temperature and duration; therefore, these four parameters are employed as input data. The output data are the microhardness values corresponding to the first stage homogenization process.

The 126 experimental data sets were combined into a single data set, and the data were randomly assigned to the training set, validation set, and test set in proportions of 70%, 15%, and 15%, respectively. Figure 4 shows the structure of the artificial neural network, which is a single hidden layer network with 10 hidden layer neurons. The network employed the Levenberg-Marquardt backpropagation algorithm. Figure 4b shows the training correlation coefficient (Training R), validation correlation coefficient (Validation R), and test correlation coefficient (Test R). The training and validation correlation coefficients are close to 0.98, suggesting that the network is well-built, and the test correlation coefficient is 0.96, indicating that the network makes accurate predictions for fresh data and has a great generalization capacity. The network would then be used to estimate the microhardness of the 5182-Sc-Zr alloy at temperatures ranging from 250 °C to 350 °C.

The predicted microhardness of the 5182-Sc-Zr alloy by the artificial neural network is shown in Figure 5. At 260 °C, 275 °C, and 290 °C treatments, the microhardness of the alloy 5182-0.05Sc-0.1Zr increased continuously without reaching stability, and it stabilized after a slight increase at 310 °C, 325 °C, and 340 °C treatments. After 35 h of 260 °C annealing, the microhardness of the alloy 5182-0.1Sc-0.1Zr became stable, and it stopped altering after 20 h at 275 °C and 290 °C. The microhardness of the 5182-0.1Sc-0.1Zr alloy declined after the homogenization process above 310 °C (including 310 °C). According to the predicted results, the microhardness of 5182-0.05Sc-0.1Zr alloy can be stabilized in the range 310 °C to 340 °C, while that of 5182-0.1Sc-0.1Zr alloy can be stabilized in the range 260 °C to 290 °C. When selecting the test temperature, the predicted findings for both alloys must be considered. The average temperature of the two temperature ranges (275 °C and 325 °C) was chosen to analyze the condition of the alloy in the next investigations, since the predicted microhardness values exhibited identical trends in each of these two temperature ranges. Figure 6 shows the microhardness and conductivity of the 5182-Sc-Zr alloy after homogenization at 275 °C and 325 °C. At the first stage of homogenization annealing at 275 °C, it was discovered that the actual microhardness of the 5182-Sc-Zr alloy was remarkably similar to the ANN prediction, with a correlation coefficient of 0.91. The correlation coefficient was 0.86 at 325 °C. With 126 sets of data modeled, the results show that the artificial neural network was able to predict the microhardness of the 5182-Sc-Zr alloy properly. Therefore, it is feasible to determine optimal parameters for the first-stage homogenization treatment based on the given experimental results without further ANN predictions or first-stage homogenization experiments.

When homogenized at 275 °C, 300 °C, 325 °C, and 350 °C, the microhardness and conductivity of the 5182-Sc-Zr alloy reached a stable state, indicating that the Al_3_Sc dispersoids had entirely precipitated. According to Shen [8], the Al_3_Sc dispersion tends to coarsen in the temperature range 300 °C to 500 °C, and to avoid coarsening of the Al_3_Sc during the first stage of homogenization annealing, the first stage of homogenization annealing of the 5182-Sc-Zr alloy should not be performed at 325 °C or 350 °C. Microhardness and conductivity can stabilize more quickly at 300 °C than at 275 °C, revealing that, the higher the temperature, the faster the Sc-containing dispersoids precipitate in the alloy. Previous studies have found that the inhomogeneous distribution of the dispersion is caused by the increase in temperature [26]. As a result, the temperature for the first stage of homogenization annealing in this study was eventually set at 275 °C. Furthermore, because the microhardness and conductivity of the 5182-Sc-Zr alloy achieved a stable state after 20 h at this temperature, 275 °C—20 h was chosen as the parameter for the first stage homogenization.

The Zr-containing supersaturated solid solution in the alloy dissolves during the second stage of homogenization annealing treatment, and Zr atoms precipitate out of the Al_3_Sc to create the Al_3_(Sc_x_Zr_1−x_) dispersoids. The microhardness and electrical conductivity of the 5182-Sc-Zr alloy are further impacted by the precipitation of the Zr-containing phase.

Figure 7 shows the variations in conductivity during the second stage homogenization. The electrical conductivity of the alloy at this stage is reduced by the dissolution of Mg atoms, while the precipitation of the Zr-containing phase can improve it [27]. During the initial 2 h of annealing at 400 °C, the conductivity of the 5182-Sc-Zr alloy decreases; however, it subsequently increases and remains unstable for up to 12 h. The change in conductivity suggests that Zr precipitates after 2 h of heating and remains unstable even after 12 h of precipitation. When subjected to homogenization at 440 °C and 480 °C, the conductivity of 5182-Sc-Zr alloys steadily increased at the start of heating and eventually remained stable at around 12 h. This phenomenon suggests the complete precipitation of Al_3_(Sc_x_Zr_1−x_) dispersoids.

To aid in the determination of the necessity for further second-stage homogenization, an artificial neural network can be developed. This network is designed to forecast the conductivity values at various temperatures (ranging from 400 °C to 480 °C). According to the research, the key factors affecting the conductivity of the alloy include the addition of Sc and Zr, and homogenization parameters such as temperature and time. As such, these four parameters are utilized as inputs for the artificial neural network model, while the resultant conductivity serves as the output data. The 72 experimental data sets were combined into a single data set, and the data were randomly assigned to the training set, validation set, and test set in proportions of 70%, 15%, and 15%, respectively. The artificial neural network, which is developed based on the parameters of the training and validation sets, is presented in Figure 8. Specifically, a single hidden layer network consisting of ten neurons in the hidden layer is utilized. Figure 8b shows the training correlation coefficients, validation correlation coefficients, and test correlation coefficients. The training and validation correlation coefficients are both greater than 0.90, indicating that the network has been effectively built, and the test correlation coefficient is greater than 0.98, indicating that the network has significant generalization capacity. As a result, the network can now be used to forecast the conductivity of the 5182-Sc-Zr alloy at temperatures ranging from 400 °C to 480 °C.

The conductivity of the 5182-Sc-Zr alloy was anticipated using the well-established network in Figure 9 after homogenization at 410 °C, 420 °C, 430 °C, 450 °C, 460 °C, and 470 °C. Figure 9 shows the projected outcomes. When the 5182-Sc-Zr alloy was homogenized at 410 °C and 430 °C, the conductivity began to grow after 2 h of holding time, and it was still not totally stable after 12 h. As a result, temperatures less than 430 °C (including 430 °C) were omitted from the subsequent second-stage homogenization experiment selection. The conductivity of the 5182-Sc-Zr alloy started to increase at the start of the annealing stage and reached stability after 8 to12 h when homogenized at 450 °C, 460 °C, and 470 °C. To confirm precision of the results, homogenization at 460 °C was used.

Figure 10 shows the microhardness and conductivity of the 5182-Sc-Zr alloy after homogenization at 460 °C. The artificial neural network successfully predicted the conductivity of the 5182-Sc-Zr alloy based on the correlation coefficient between the measured and predicted conductivity values, which was 0.93. The available second-stage homogenization treatment temperatures are 440 °C, 460 °C, and 480 °C. When kept at 480 °C, the microhardness of the 5182-0.1Sc-0.1Zr alloy decreased more noticeably, demonstrating that the strengthening effect of the Al_3_(Sc_x_Zr_1−x_) was influenced; hence, 480 °C was ruled out as the second-stage temperature. The trend of alloy properties was the same when homogenized at 440 °C and 460 °C and, considering the influence of temperature on the coarsening of the dispersoids, 440 °C was finally chosen as the temperature for the second stage homogenization. Based on the changes in microhardness and conductivity, it was determined that the alloy’s properties reached stability at 440 °C for 12 h. The parameters of the second-stage homogenizing annealing were determined to be 440 °C for 12 h. ANNs have been effectively utilized in this study to predict data and choose experimental parameters. Based on the ANN prediction results, the settings for the first stage of homogeneous annealing treatment (275 °C for 20 h) and the second stage (440 °C for 12 h) were chosen.

### 3.2. Microstructure Evolution of 5182-Sc-Zr Alloy during Homogenization

Figure 11 shows the microstructure of 5182-Sc-Zr alloy before and after homogenization. The average grain size of the 5182-0.1Sc-0.1Zr alloy ingots is 0.2 mm (±0.1 mm), while that of the 5182 alloy ingots is 0.4 mm (±0.2 mm). The metallographic characteristics of 5182-0.1Sc-0.1Zr alloy did not change when homogenization temperatures varied. Note the average grain size of as-homogenized 5182-0.1Sc-0.1Zr alloy, which was measured to be 0.2 mm (±0.1 mm), in consent with that of the ingots.

The EDS mapping of ingots and as-homogenized 5182-0.1Sc-0.1Zr alloy is shown in Figure 12. During casting solidification, Mg atoms do not have enough time to disperse, leading to non-equilibrium crystallization and the formation of non-equilibrium structures [28]. Figure 12 illustrates the segregation of magnesium in the ingots. After two-stage homogenization, Mg atom diffusion in the 5182-Sc-Zr alloy corrects elemental segregation. In addition to EDS mapping of the elements, properties of alloy microregions can also reveal the reduction of segregation. The alloy’s microhardness can be employed in this study as an indication of the elemental segregation degree.

The alloy microhardness was determined by measuring 10 hardness values for each sample at 2 mm intervals. We use standard deviations to determine fluctuations in microhardness values, and large fluctuations indicate that the sample is not homogeneous in microstructure, i.e., there is elemental segregation in the alloy. Figure 13 shows the standard deviation of the 5182-Sc-Zr alloy microhardness after homogenization. The present study demonstrated that the uniformity of microhardness in 5182-Sc-Zr was improved via homogenization. Specifically, the standard deviation of microhardness for AA5182 decreased from 3.8 to 1.4, that of 5182-0.05 Sc-0.1Zr decreased from 2.1 to 1.4, and that of 5182-0.1Sc-0.1Zr decreased from 3.6 to 1.7. These findings suggest that element segregation was mitigated as a result of the homogenization annealing process.

Figure 14 shows TEM images of the 5182-0.05Sc-0.1Zr alloy and the 5182-0.1Sc-0.1Zr alloy after two-stage homogenization treatment. In Figure 14a, no dispersoids were found because of the low Sc concentration and tiny size of the Al_3_Sc dispersoids. The Al_3_Sc dispersoids can be observed in the 5182-0.1Sc-0.1Zr alloy that has undergone the first stage homogenization. The dispersoids were detected in both 5182-0.05Sc-0.1Zr alloy and 5182-0.1Sc-0.1Zr alloy after the second stage homogenization, as illustrated in Figure 14c,d.

Figure 15 displays particle size statistics of the Al_3_(Sc_x_Zr_1−x_)dispersoids after a two-stage homogenization treatment. The average diameters of the Al_3_(Sc_x_Zr_1−x_) dispersoids in the 5182-0.05Sc-0.1Zr alloy and the 5182-0.1Sc-0.1Zr alloy were 9.6 nm and 5.1 nm, respectively, with the maximum diameter reaching 15 nm.

The size variation of the dispersoids may reveal the relationship between the dispersoids and the matrix. The Al_3_(Sc_x_Zr_1−x_) dispersoids are coherent with the Al matrix at room temperature [29]. During the subsequent processing, Al_3_(Sc_x_Zr_1−x_) obtained by double-stage homogenization annealing will undergo thermal processes, such as thermal deformation and recrystallization annealing, which will coarsen the dispersoids and increase lattice mismatch between the coarsened dispersoids and the Al matrix. The coherent interaction between the Al_3_(Sc_x_Zr_1−x_) dispersoids and the Al matrix will be destroyed if the lattice mismatch exceeds the Burgers vector, resulting in the formation of interfacial dislocations and a significantly reduced strengthening impact of the dispersoids. The relationship between the Al_3_(Sc_x_Zr_1−x_) dispersoids and the Al matrix after homogenization must be recognized to avoid excessive coarsening. Equation (1) can be used to calculate theoretical critical radius of the coherent relationship [30]:r = b/2θ,(1)
where r is the theoretical critical radius, b is the dislocation Burgers vector (0.286 nm), and θ is the misfit between the Al_3_(Sc_x_Zr_1−x_) dispersoids and the Al matrix (1.25%) [30]. According to Equation (1), the theoretical critical diameter was 22.8 nm. Comparing the theoretical critical size with the statistics of dispersoids sizes, it was found that the Al_3_(Sc_x_Zr_1−x_) dispersoids remained coherent with the Al matrix after the second stage homogenization.

The aforementioned comparison between the size of dispersions and their theoretical critical size lacks experimental substantiation. The diffraction spots corresponding to the (200) crystallographic planes were selected for inverse Fourier transformation via high-resolution transmission micro-images, enabling the determination of the interplanar spacing. As illustrated in Figure 16, the interplanar spacing of the (200) crystallographic planes of the Al matrix was measured to be 0.18 nm. Likewise, after the homogenization treatment, the (200) crystallographic planes of the Al_3_(Sc_x_Zr_1−x_) dispersoids with a diameter of 15 nm also exhibited an interplanar spacing of 0.18 nm, indicating that the Al_3_(Sc_x_Zr_1−x_) dispersoids maintained a desirable coherency with the matrix after two-stage homogenization.

### 3.3. Room Temperature Mechanical Properties and Microstructure of 5182-Sc-Zr Alloy after Recrystallization Annealing

The 5182-Sc-Zr alloy forms nanoscale Al_3_(Sc_x_Zr_1−x_) dispersoids after undergoing a two-stage homogenization treatment. This phase exhibits excellent capacity to prevent recrystallization during the recrystallization annealing process [31,32]. Moreover, even with trace amounts of Sc and Zr, the microstructure and mechanical properties of the aluminum alloy can be significantly altered at room temperature [33,34,35,36].

Samples with 70% cold rolling reduction were subjected to annealing at 250 °C, 300 °C, 400 °C, and 500 °C for a duration of 2 h, with the explicit intention of investigating the impact of Al_3_(Sc_x_Zr_1−x_) dispersoids on the mechanical properties of the 5182-Sc-Zr alloy. The stress–strain curves for cold-rolled sheet and annealed sheet of the 5182-Sc-Zr alloy are shown in Figure 17. The findings revealed that a positive correlation existed between Sc content and the yield strength of 5182-Sc-Zr alloy. The strength of the 5182-0.1Sc-0.1Zr alloy is significantly higher than the other two alloys under cold rolling conditions. Upon recrystallization annealing at 250 °C, the strength differential between the 5182-0.05Sc-0.1Zr alloy and the 5182-0.1Sc-0.1Zr alloy was found to decrease relative to their cold-rolled state. With the further increase in recrystallization temperature, there are significant strengthening effects due to the abundant deformation structure and Al_3_(Sc_x_Zr_1−x_) dispersoids present in the 5182-0.05Sc-0.1Zr alloy and the 5182-0.1Sc-0.1Zr alloy.

The microstructure was assessed through electron backscatter diffraction (EBSD) analysis. Longitudinal sections were utilized for all images, and the resulting microstructures have been depicted in Figure 18, Figure 19 and Figure 20. In Figure 18, the 5182 alloy underwent complete recrystallization after annealing at 300 °C, and abnormal growth grains were already evident at 500 °C. In Figure 19 and Figure 20, the annealed 5182-0.05Sc-0.1Zr alloy and the 5182-0.1Sc-0.1Zr alloy retain a large amount of deformed microstructure. After recrystallization annealing at 250 °C, no recrystallized grains were found in the 5182-Sc-Zr alloy. After annealing at 300 °C, a modest number of finely recrystallized grains occurred in the 5182-0.05Sc-0.1Zr alloy. As the annealing temperature increased, the recrystallized grains became increasingly visible in the microstructure. Comparing the recrystallization degrees of the alloys in Figure 18, Figure 19 and Figure 20, it can be found that the recrystallization degree of the 5182-Sc-Zr alloy decreases with the increase of the Sc content at the same recrystallization temperature. The recrystallization rates of 5182-0.05Sc-0.1Zr alloy and 5182-0.1Sc-0.1Zr alloy were determined to be 15.5% and 15.7%, respectively, after annealing at 300 °C, whereas complete recrystallization was observed in 5182 alloy. Notably, a mere 36.4% and 30.1% recrystallization rates were achieved following 2 h of annealing at 500 °C for 5182-0.05Sc-0.1Zr alloy and 5182-0.1Sc-0.1Zr alloy, respectively.

The microstructure of the 5182-0.1Sc-0.1Zr alloy was further investigated by TEM. Figure 21 shows TEM bright field images of the 5182-Sc-Zr alloy. After a 2-h annealing treatment at 250 °C and 300 °C, the grains and sub-grains of the 5182-0.1Sc-0.1Zr alloy maintain their elongated fibrous morphology along the rolling direction. Notably, copious entangled dislocations and diffuse Al_3_(Sc_x_Zr_1−x_) dispersoids are observed.

Based on the above phenomenon, it can be concluded that the presence of the Al_3_(Sc_x_Zr_1−x_) dispersoids has a significant impact on the microstructure of the 5182-Sc-Zr alloy. During recrystallization annealing, dislocations, sub-grains, and grain boundaries are pinned by the Al_3_(Sc_x_Zr_1−x_) dispersoids, which inhibit recrystallization of the 5182-Sc-Zr alloy. As a result, the alloy retains a large amount of its deformed microstructure after annealing. These deformed structures and dispersoids will affect the mechanical properties of the alloy [37].

The results showed that, as the annealing temperature increased, the mechanical stability of the alloy 5182-Sc-Zr improved compared to that of the 5182 alloy. This phenomenon can be attributed to the Zener pinning effect of the dispersoids on dislocation motion [38]. During cold rolling, the motion dislocation was hindered, resulting in the formation of substructures in the 5182-Sc-Zr alloy. The cold-rolled 5182-Sc-Zr alloy showed an increase in strength through a combination of substructure strengthening and dispersoids strengthening during tensile testing. After annealing at 250 °C, static recovery occurred in the 5182-Sc-Zr alloy, reducing the effectiveness of substructure strengthening. This is due to the substructures formed during cold deformation being in a metastable state, and high-temperature annealing of the 5182-Sc-Zr alloy resulted in static recovery of the partially unstable substructures, leading to a reduction in alloy strength. Meanwhile, the stable substructures were preserved due to the pinning effect of the Al_3_(Sc_x_Zr_1−x_) dispersoids, and these substructures enhanced the strength of the Sc and Zr bearing alloys during tensile testing. The results showed that the presence of Al_3_(Sc_x_Zr_1−x_) dispersoids had a significant pinning effect on the 5182-Sc-Zr alloy during annealing, preventing recrystallization and hindering dislocation motion during subsequent tensile deformation, thereby improving the microstructural stability and mechanical stability of the alloy 5182-Sc-Zr.

The coherency relationship between the Al_3_(Sc_x_Zr_1−x_) dispersoids and the matrix is influenced by the size of the dispersoids. Once the dispersoids exceed a certain size, they lose their coherency with the matrix, and this size is referred to as the critical size for the coherency-to-incoherency transition. When the size of dispersoids is smaller than the critical size, the dispersoids coherently strengthen the matrix via both coherency strengthening and modulus mismatch strengthening. However, when the size of dispersoids is larger than the critical size, Orowan strengthening takes place. The critical size that influences the strengthening mechanism has certain variations among different alloys. According to Marquis’ research, when the size of dispersoids exceeds 2.4 nm, the dislocation motion mechanism should be the bypass one, resulting in Orowan strengthening [39]. Vo conducted a study to investigate the effect of precipitate diameter on the dispersoid strengthening of the Al-0.055Sc-0.005Er-0.02Z-0.05Si alloy and observed that the dispersoid strengthening decreased from 97 MPa to 62 MPa as the precipitate diameter increased from 8 nm to 12 nm, following an inverse proportionality with the size of the dispersoid in accordance with the Orowan strengthening law [40]. These results suggest that the critical size for the transition of the dispersoid strengthening mechanism in the studied Al-0.055Sc-0.005Er-0.02Z-0.05Si alloy is less than 8 nm. Another study indicates that the critical size for the transition of the dispersoid strengthening mechanism should be around 10 nm [41]. The theoretical critical diameter for coherency transition obtained in previous studies is 22.8 nm. In the previous section, it has been observed that there are many dispersoids existing in the microstructure of the alloy. Statistical analysis of the Al_3_(Sc_x_Zr_1−x_) dispersoids size was further conducted. The size of Al_3_(Sc_x_Zr_1−x_) dispersoids in the annealed alloy were obtained by transmission electron microscopy, as shown in Table 2.

Statistical results show that the average size of Al_3_(Sc_x_Zr_1−x_) dispersoids in the 5182-0.1Sc-0.1Zr alloy increases with the increase of recrystallization annealing temperature. After annealing at 250 °C and 300 °C, the size of dispersoids did not reach the theoretical coherency transition critical size. After annealing at 400 °C, some dispersoids approached the critical size, while after annealing at 500 °C, the average size of Al_3_(Sc_x_Zr_1−x_) dispersoids exceeded the theoretical critical size and reached 25 nm. Therefore, we selected 25 nm and 50 nm Al_3_(Sc_x_Zr_1−x_) dispersoids after annealing at 400 °C and 500 °C, respectively, to observe their coherency relationship with the Al matrix. Figure 22 shows HRTEM images of Al_3_(Sc_x_Zr_1−x_) dispersoids. The interplanar spacing was measured by selecting the corresponding diffraction spots of the (200) plane and performing Fourier transform on the obtained images. It was found that the interplanar spacing of the aluminum matrix (200) plane was 0.18 nm. After annealing at 400 °C, the interplanar spacing of the (200) plane of both the Al_3_(Sc_x_Zr_1−x_) dispersoids and the interface was also 0.18 nm, confirming that the 25 nm Al_3_(Sc_x_Zr_1−x_) dispersoid had a good coherency relationship with the matrix. After annealing at 500 °C, the interplanar spacing of the interface and Al_3_(Sc_x_Zr_1−x_) dispersoid (200) plane increased to 0.21 nm, indicating that the misfit between Al_3_(Sc_x_Zr_1−x_) dispersoids and the matrix increased. Based on the analysis of the coherency relationship between Al_3_(Sc_x_Zr_1−x_) dispersoids and the matrix, it can be determined that, after recrystallization annealing at a temperature lower than 400 °C (including 400 °C), the Al_3_(Sc_x_Zr_1−x_) dispersoids in the 5182-Sc-Zr alloy mainly contribute to the strengthening mechanism by coherency strengthening and modulus mismatch strengthening. After recrystallization annealing at 500 °C, as the coherency relationship between some Al_3_(Sc_x_Zr_1−x_) dispersoids and the matrix changed, the strengthening mechanism of the dispersoids was no longer completely dominated by coherency strengthening and modulus mismatch strengthening, but under the semi-coherent state, the strengthening mechanism of the 5182-Sc-Zr alloy would not completely transform to Orowan strengthening.

### 3.4. Establishment of ANN for Room Temperature Performance of 5182-Sc-Zr Alloy

Previously, the development of artificial neural networks for alloy property prediction only took into account the impact of individual processing or heat treatment conditions on alloy properties [11,12,42,43,44]. However, limited attention was given to models that considered the collective influence of multiple processing or heat treatment process parameters on the ultimate properties of the alloys. In this study, the effects of homogenization and recrystallization annealing treatments on the characteristics of the 5182-Sc-Zr alloy are investigated. The formation of Al_3_(Sc_x_Zr_1−x_) dispersoids during homogenization treatment effectively impedes recrystallization of the alloy during subsequent annealing processes, while the preserved substructure in the alloy collaborates with the Al_3_(Sc_x_Zr_1−x_) dispersoids to bolster the strength of the 5182-Sc-Zr alloy. Since the strengthening effect of the Al_3_(Sc_x_Zr_1−x_) dispersoids on the alloy is contingent on the state of the dispersoids, and the latter can be regulated by the homogenization treatment, the parameters of this treatment not only condition the properties of the alloy during homogenization, but also exert a persistent impact on its characteristics during subsequent processing and heat treatment. The ability of the artificial neural network to deal with this multi-threaded nonlinear relationship can then be fully utilized to obtain a more comprehensive model covering multiple phases of influencing factors by introducing the parameters of homogenization and recrystallization annealing together as influencing factors, when building an artificial neural network for room temperature property prediction of 5182-Sc-Zr alloy.

The artificial neural network employed in this study is an error back-propagation artificial neural network (BP-ANN), which determines the smallest error between the output data and the target data. To avoid “local minima” in the parameters, we further improve the model through Genetic Algorithms (GA) [45]. Finally, a cross-validation approach is performed to determine the model’s generalization capabilities.

To construct an artificial neural network for forecasting the room temperature properties of 5182-Sc-Zr alloy, various input data have been employed, which encompass the parameters of the first-stage homogenization temperature T_1_ (°C) and time t_1_ (h), the second-stage homogenization temperature T_2_ (°C) and time t_2_ (h), the cold deformation strain ε (%), the recrystallization annealing temperature T_R_ (°C), and the Sc and Zr contents (wt.%). The output data of the system comprises the corresponding stress, denoted by the symbol σ (MPa). Figure 23 shows the construction of the artificial neural network used to forecast the yield stress of the 5182-Sc-Zr alloy at room temperature.

The connection weights between the input layer and the hidden layer in the artificial neural network construction for predicting the room temperature mechanical properties of 5182-Sc-Zr alloy successively affect the input data, output data, and finally the error between the experimental data and the predicted data in the output layer. After adjusting the connection weights between neurons and neuron thresholds in accordance with the mistakes in reverse, an artificial neural network that can forecast the yield stress of 5182-Sc-Zr alloy at room temperature is produced. The network is a single hidden-layer network with 25 neurons. Figure 24 shows the comparison of the predicted data obtained by the network with the experimental data, the correlation coefficient (R), and the average absolute relative error (AARE). As shown in Figure 24a, the predicted values generated by artificial neural network exhibit remarkable coincide with the original stress–strain curve. As evidenced in Figure 24b, the correlation coefficient between the predicted data and experimental data equals 0.98 and the AARE is negligible. Thus, based on these observations, it can be determined that the deployed artificial neural network possesses exceptional precision.

To achieve further optimization of the artificial neural network, and to overcome the challenge of network parameters being trapped in a localized minimum, a genetic algorithm (GA) will be deployed to optimize the existing network structure. The goal is to minimize prediction errors, leading to enhanced performance of the network model. The prediction results of GA-ANN and corresponding correlation coefficients, and average absolute relative errors are shown in Figure 25. After optimization by genetic algorithm, the predicted data and the experimental data are almost identical. Compared with the errors of the artificial neural network model, the correlation coefficient has improved from 0.98 to 0.99, and the average absolute relative error has decreased from 1.39% to 0.52%. Based on the changes in the correlation coefficient and the average absolute relative error, it can be determined that the accuracy of the artificial neural network for predicting the room temperature mechanical properties of the 5182-Sc-Zr alloy has been enhanced.

The generalization ability of the network is considered while examining the application of artificial neural networks. The generalization ability of a network is typically assessed by evaluating its performance on previously unseen data, which are not included in the training dataset. Specifically, the network is constructed using dataset D, and then new input parameters that are not present in D are presented to the network. When the resulting prediction is accompanied by a small discrepancy with the corresponding experimental observations, the network is regarded as highly generalizable, whereas otherwise, it is deemed poorly generalizable. Upon establishing an artificial neural network to forecast the mechanical properties of 5182-Sc-Zr alloy at room temperature, all available experimental data were incorporated into the training dataset. Due to approximating the training error of the training set as the generalization error, it remains uncertain whether the network is capable of accurately predicting parameters other than those in the training set.

To test the generalization ability of the artificial neural network, the cross-validation method was used. When constructing the network, the total data set of 198 data sets was sampled into four sub-data sets. Three data sets were utilized to train the network at the same time, and one data set was used to assess the generalization error of the network. To properly test the network, the four sub-datasets are mutually exclusive. Following the completion of four separate rounds of training, the average outcome may be obtained and analyzed. As presented in Figure 26, the test results reveal an excellent linear fit to the training set data, with a correlation coefficient of 0.98 and an average absolute relative error of 1.79%. While some elements of the test set data deviated substantially from the linear relationship, leading to a slight decline in the correlation coefficient between predicted and experimental data, the majority of the predicted data still exhibit a strong linear relationship with the experimental data. These observations demonstrate that the artificial neural network possesses the capacity to generate precise predictions for elements of the test set which had not been included in the training process.

## 4. Conclusions

The precipitation of the Al_3_(Sc_x_Zr_1−x_) dispersoids and its effect on the room temperature mechanical properties and microstructure of the 5182-Sc-Zr alloy are studied in this paper. A model for selecting parameters and a network for forecasting alloy property are provided. By studying the precipitation of the dispersion phase of 5182-Sc-Zr alloy after homogenization annealing and the changes of microstructure and mechanical properties after recrystallization annealing, the following conclusions are obtained:An artificial neural network was constructed to model the homogenization treatment process of 5182-Sc-Zr alloy by employing a dataset that included 72 to 126 sets of experimental data. Through this process, the optimal parameters for the two-stage homogenization annealing were determined. The treatment parameters for homogenization are as follows: a temperature of 275 °C lasting 20 h for the first stage, followed by a second stage held at 440 °C for 12 h.The Al_3_(Sc_x_Zr_1−x_) dispersoids bring significant strengthening effects to the 5182-Sc-Zr alloy. Based on the coherency relationship between the dispersoids and the aluminum matrix, it can be ascertained that, subsequent to annealing at 400 °C, the dispersoids will maintain its coherency relationship with the matrix, thereby imparting coherency strengthening to the alloy. In addition to dispersoids strengthening, the fibrous microstructure in the 5182-Sc-Zr alloy brings about grain boundary strengthening, and the joint action of dispersoids and grain boundary improves the stability of the mechanical properties at high temperatures.In the development of an artificial neural network to forecast room temperature properties of 5182-Sc-Zr alloy, the parameters of both homogenization and recrystallization annealing were integrated as influential factors. This approach facilitated the creation of a model that can reflect the synergistic influence of multiple threaded factors on mechanical properties. With this network, predictions were generated with exceptional accuracy for previously unutilized datasets that were not included in the initial network training. These results further validate the model’s strong generalization capability.The predictive results of ANN for the mechanical properties of the 5182-Sc-Zr alloy were compared to the original experimental data, yielding a correlation coefficient of 0.98. The average absolute relative error was found to be 1.39%. Following the optimization by GA, the accuracy of the network was further improved, resulting in a reduction of the average absolute relative error between the predicted and experimental data to 0.52%.

## Figures and Tables

**Figure 1 materials-16-05315-f001:**
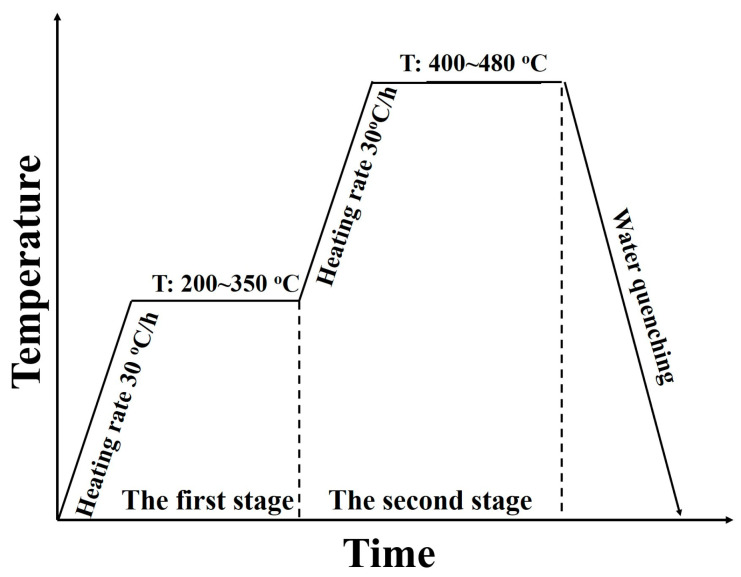
Homogenization experiment process.

**Figure 2 materials-16-05315-f002:**
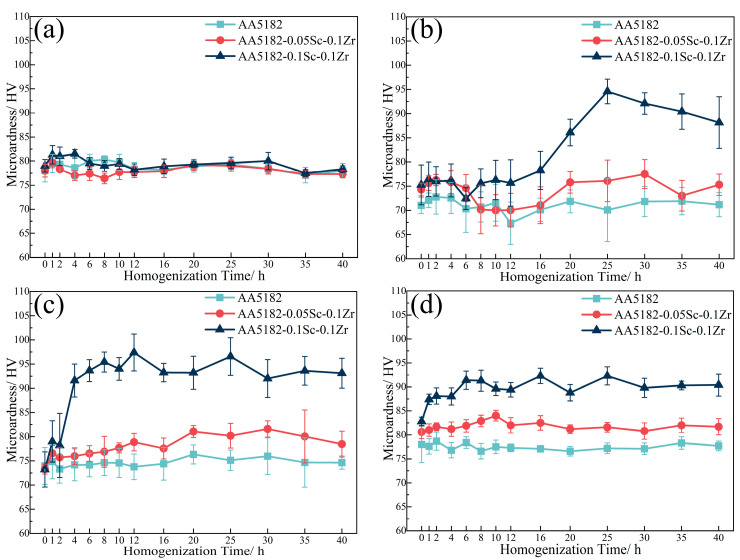
Microhardness of 5182-Sc-Zr alloy at (**a**) 200 °C, (**b**) 250 °C, (**c**) 300 °C, and (**d**) 350 °C for the first stage of homogenization annealing treatment.

**Figure 3 materials-16-05315-f003:**
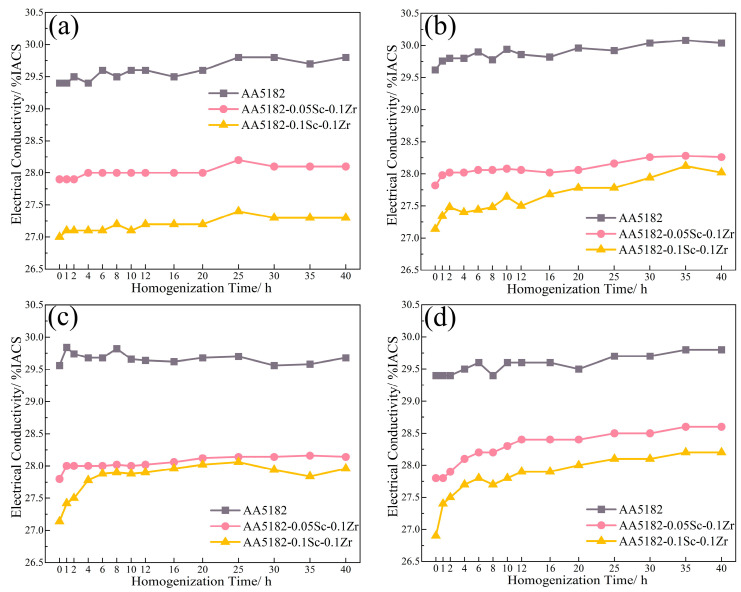
Electrical conductivity of 5182-Sc-Zr alloy at (**a**) 200 °C, (**b**) 250 °C, (**c**) 300 °C, and (**d**) 350 °C for the first stage of homogenization annealing treatment.

**Figure 4 materials-16-05315-f004:**
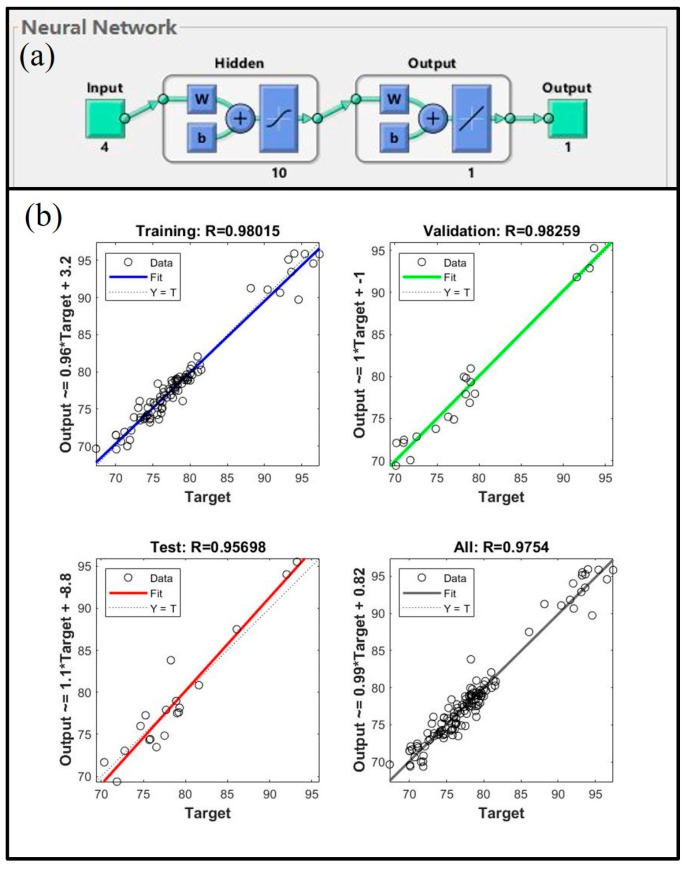
(**a**) The structure of the ANN, (**b**) the Training R, Validation R and Test R of the ANN in the temperature range of 250~350 °C during the first stage homogenization.

**Figure 5 materials-16-05315-f005:**
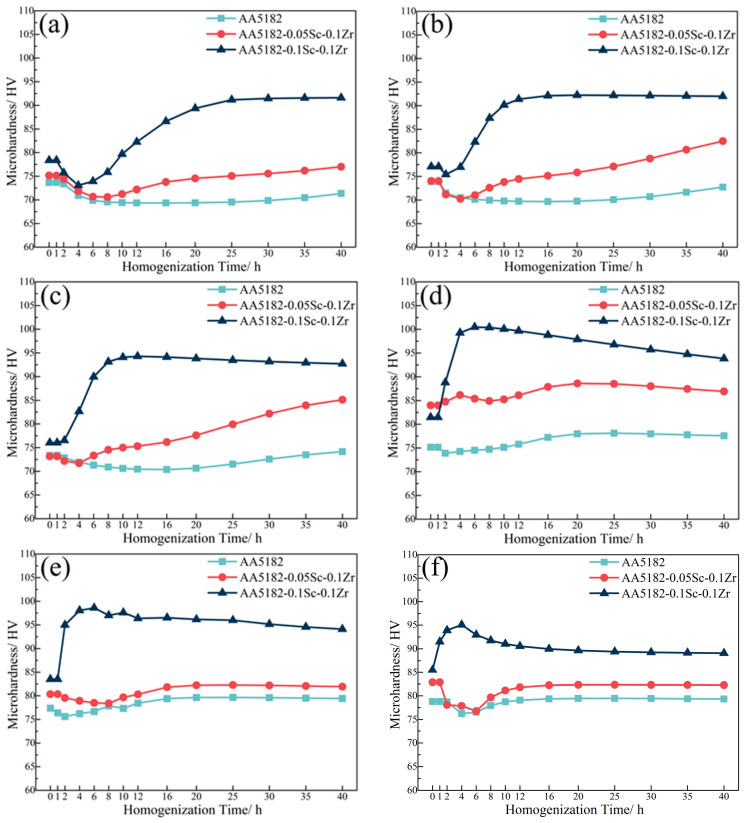
ANN predicts the microhardness of the alloy during the first stage of homogenization at (**a**) 260 °C, (**b**) 275 °C, (**c**) 290 °C, (**d**) 310 °C, (**e**) 325 °C, (**f**) 340 °C.

**Figure 6 materials-16-05315-f006:**
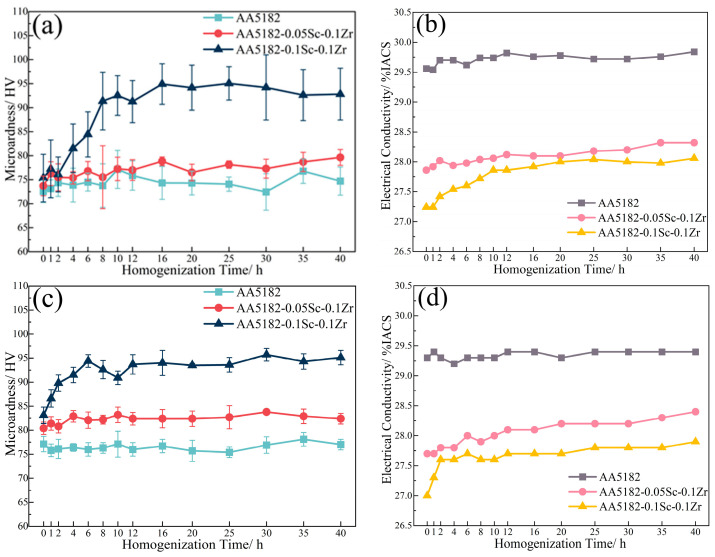
Microhardness and conductivity of the three alloys during the first stage homogenization at (**a**,**b**) 275 °C, (**c**,**d**) 325 °C.

**Figure 7 materials-16-05315-f007:**
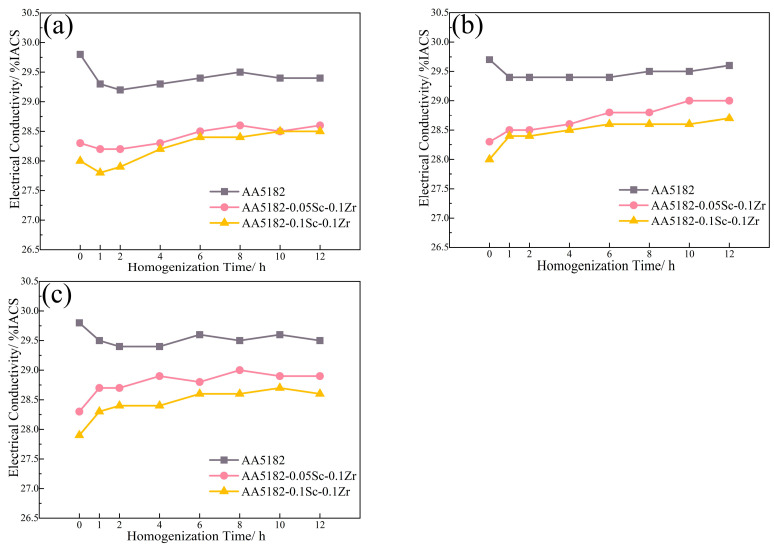
Electrical conductivity of 5182-Sc-Zr alloys during the second stage homogenization at (**a**) 400 °C, (**b**) 440 °C, (**c**) 480 °C.

**Figure 8 materials-16-05315-f008:**
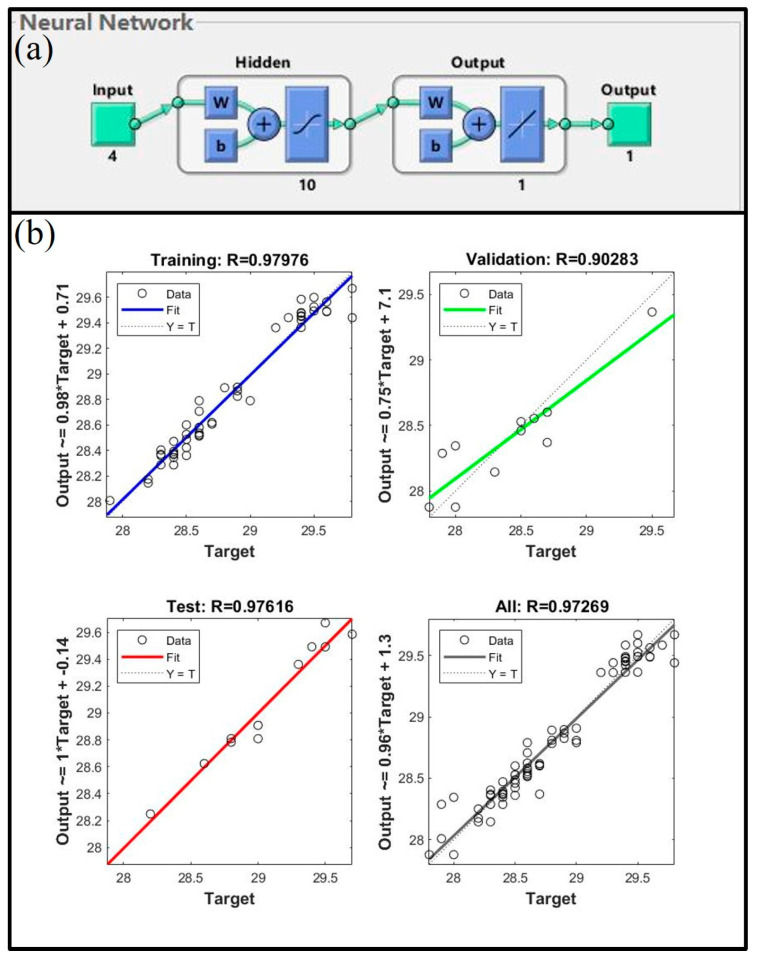
(**a**) The structure of the ANN, (**b**) the Training R, Validation R and Test R of the ANN in the temperature range 400–480 °C during the second stage homogenization.

**Figure 9 materials-16-05315-f009:**
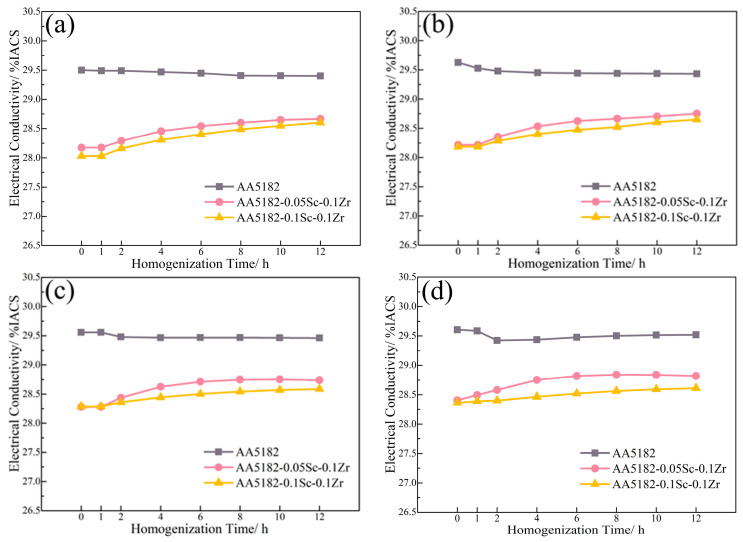
ANN predicts the conductivity of the alloy during the second stage of homogenization at (**a**) 410 °C, (**b**) 420 °C, (**c**) 430 °C, (**d**) 450 °C, (**e**) 460 °C, (**f**) 470 °C.

**Figure 10 materials-16-05315-f010:**
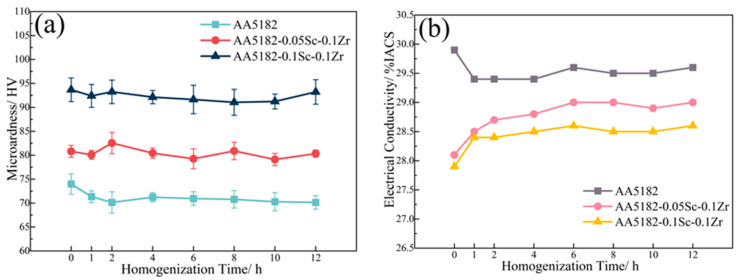
(**a**) Microhardness and (**b**) electrical conductivity of the alloys during the second stage homogenization at 460 °C.

**Figure 11 materials-16-05315-f011:**
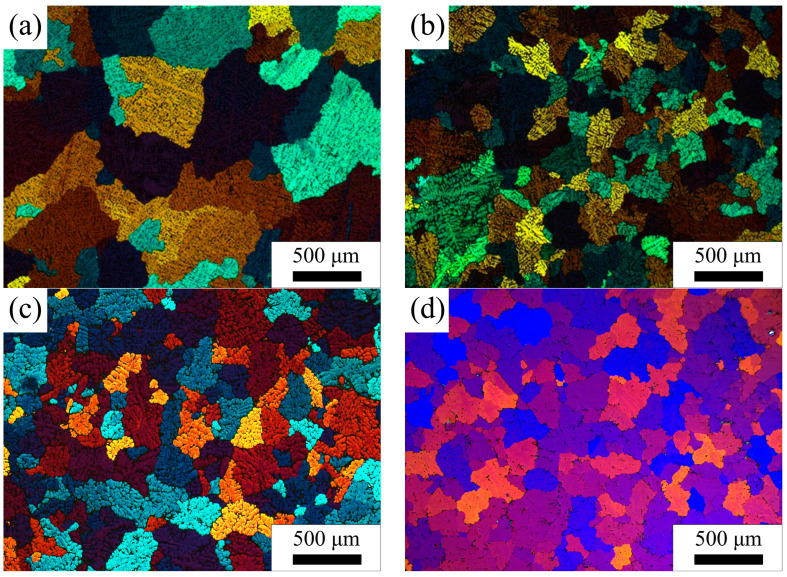
Microstructure of (**a**) AA5182 ingots, (**b**) AA5182-0.1Sc-0.1Zr ingots, (**c**) AA5182-0.1Sc-0.1Zr after the first stage homogenization at 275 °C for 20 h, (**d**) AA5182-0.1Sc-0.1Zr after the second stage homogenization at 440 °C for 12 h.

**Figure 12 materials-16-05315-f012:**
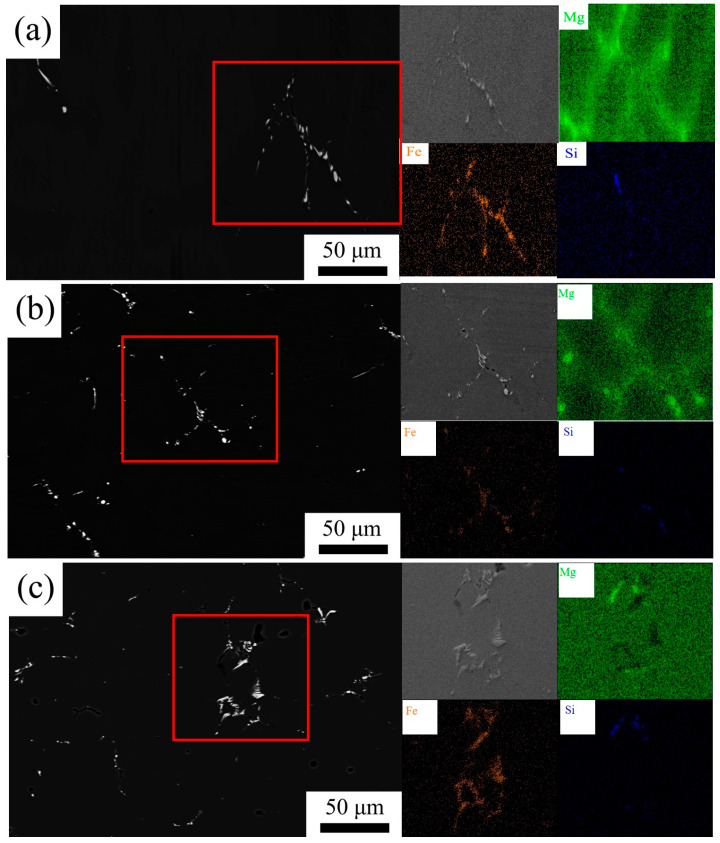
SEM-BSD images and Mg EDS map of (**a**) as-cast, (**b**) the first stage isothermal annealed, (**c**) the second stage isothermal annealed AA5182-0.1Sc-0.1Zr.

**Figure 13 materials-16-05315-f013:**
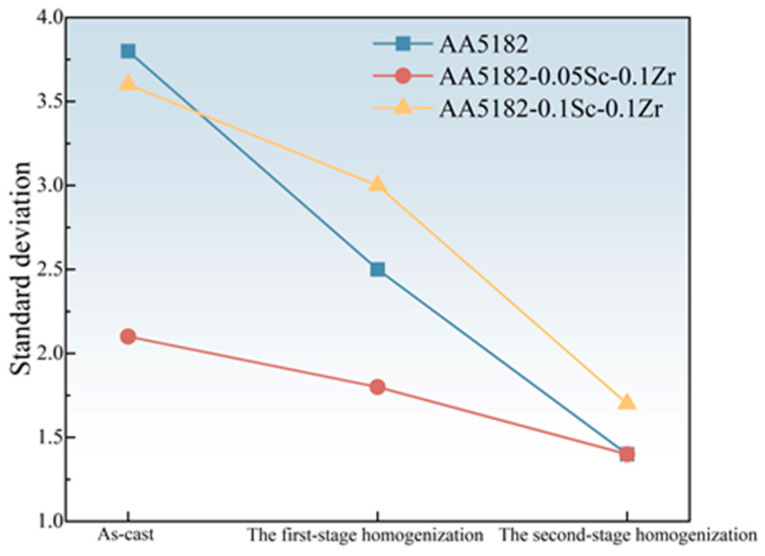
Standard deviation of microhardness after two-stage homogenization.

**Figure 14 materials-16-05315-f014:**
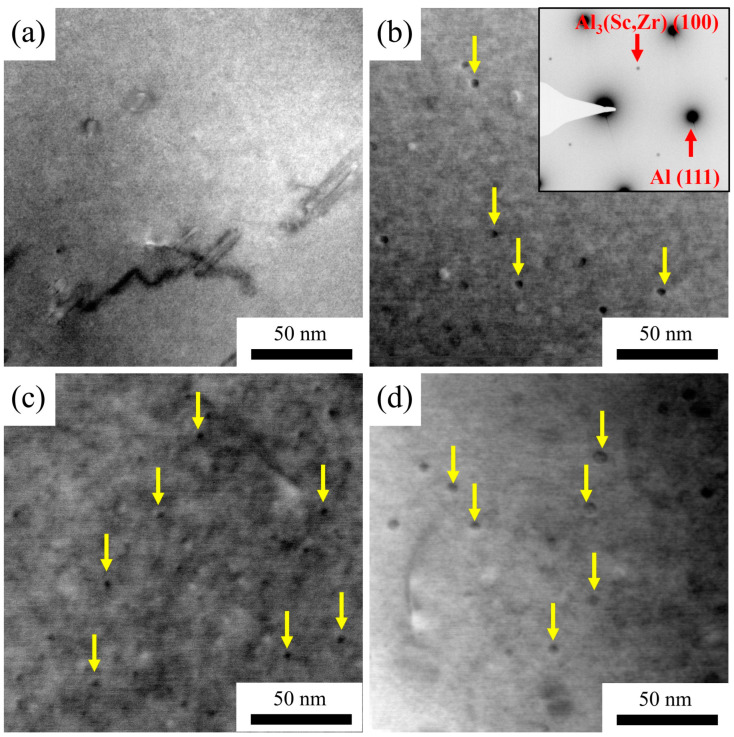
The TEM bright field images of (**a**) the first stage homogenized AA5182-0.05Sc-0.1Zr, (**b**) the second stage homogenized AA5182-0.05Sc-0.1Zr, (**c**) the first stage homogenized AA5182-0.1Sc-0.1Zr, (**d**) the second stage homogenized AA5182-0.1Sc-0.1Zr (yellow arrow points to Al_3_Sc/Al_3_(Sc_x_Zr_1−x_).

**Figure 15 materials-16-05315-f015:**
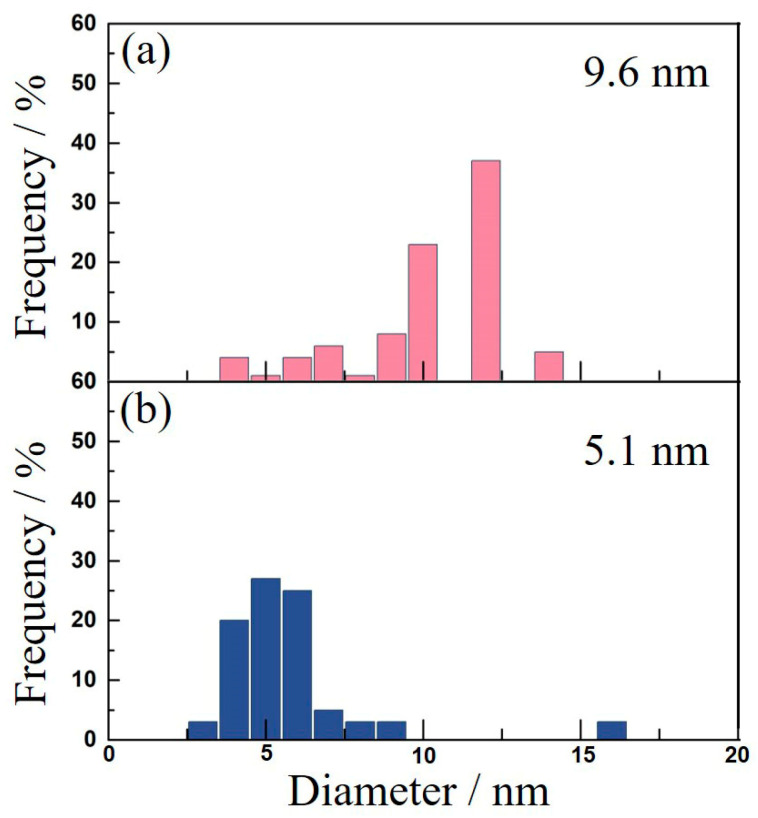
Particle size distribution of Al_3_(Sc_x_Zr_1−x_) in (**a**) AA5182-0.05Sc-0.1Zr, (**b**) AA5182-0.1Sc-0.1Zr.

**Figure 16 materials-16-05315-f016:**
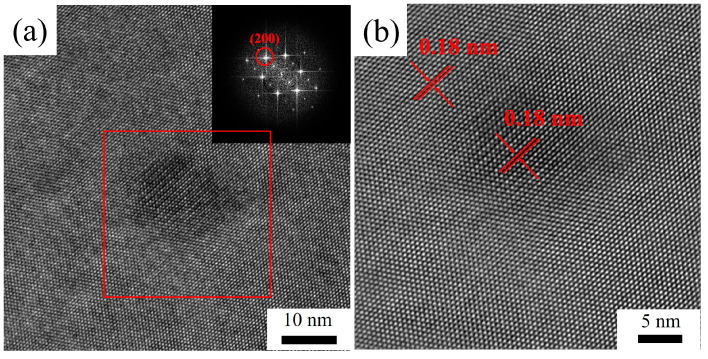
(**a**) HRTEM image after the second homogenization, (**b**) partial region of HRTEM image after the second homogenization.

**Figure 17 materials-16-05315-f017:**
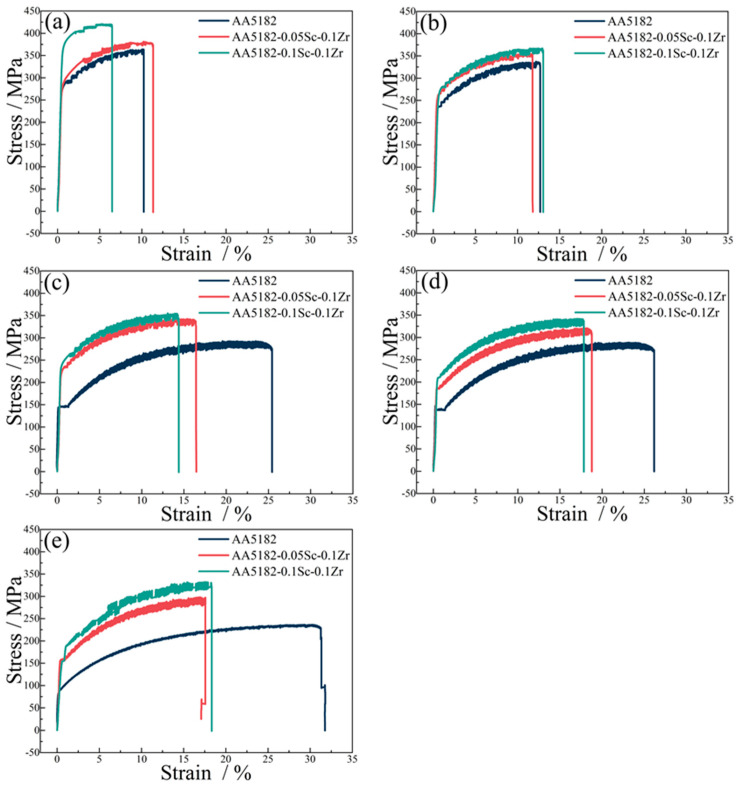
Tensile stress–strain curves of 5182-Sc-Zr alloy in (**a**) cold rolling state, (**b**) 250 °C annealing state, (**c**) 300 °C annealing state, (**d**) 400 °C annealing state, (**e**) 500 °C annealing state.

**Figure 18 materials-16-05315-f018:**
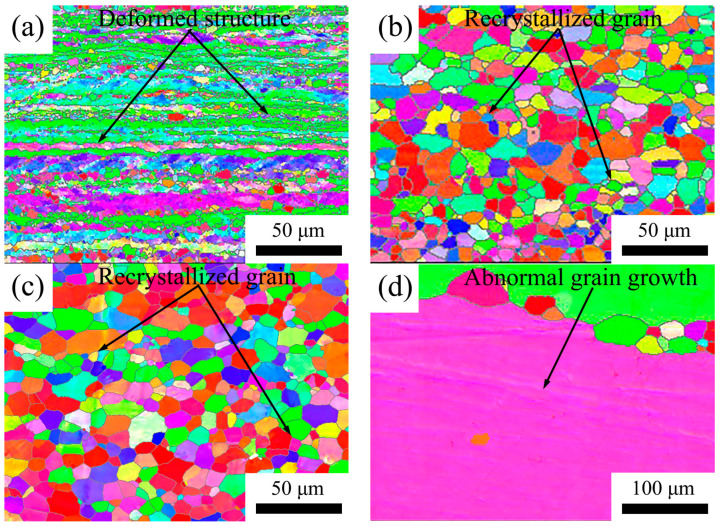
Grain orientation maps of 5182 alloy annealed at (**a**) 250 °C, (**b**) 300 °C, (**c**) 400 °C, (**d**) 500 °C.

**Figure 19 materials-16-05315-f019:**
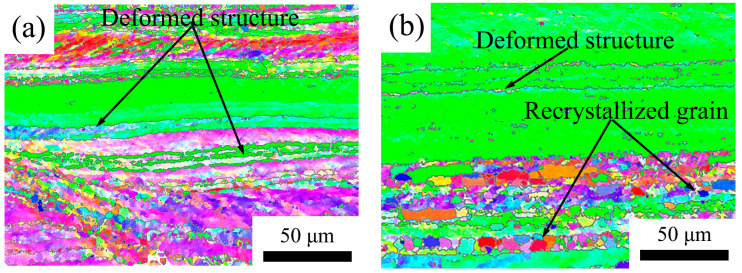
Grain orientation maps of 5182-0.05Sc-0.1Zr alloy annealed at (**a**) 250 °C, (**b**) 300 °C, (**c**) 400 °C, (**d**) 500 °C.

**Figure 20 materials-16-05315-f020:**
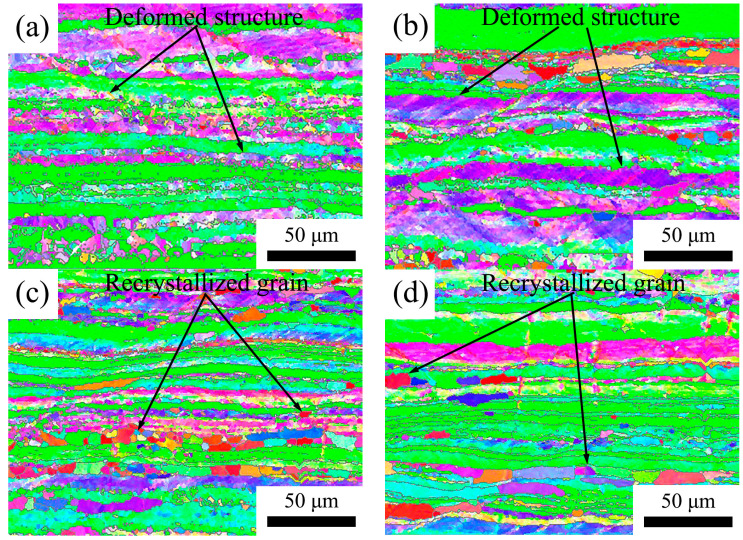
Grain orientation maps of 5182-0.1Sc-0.1Zr alloy annealed at (**a**) 250 °C, (**b**) 300 °C, (**c**) 400 °C, (**d**) 500 °C.

**Figure 21 materials-16-05315-f021:**
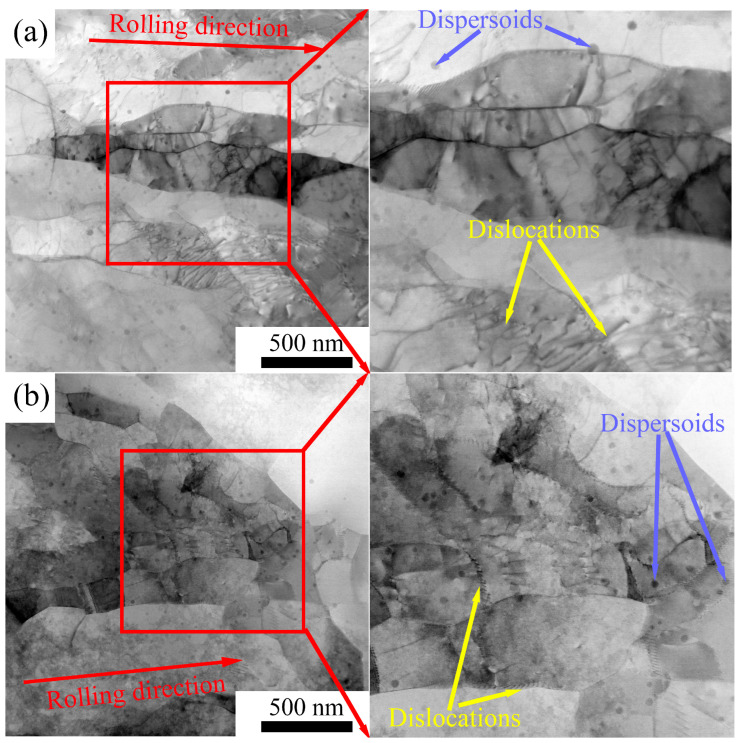
TEM bright field images of 5182-0.1Sc-0.1Zr alloy annealed at (**a**) 250 °C, (**b**) 300 °C.

**Figure 22 materials-16-05315-f022:**
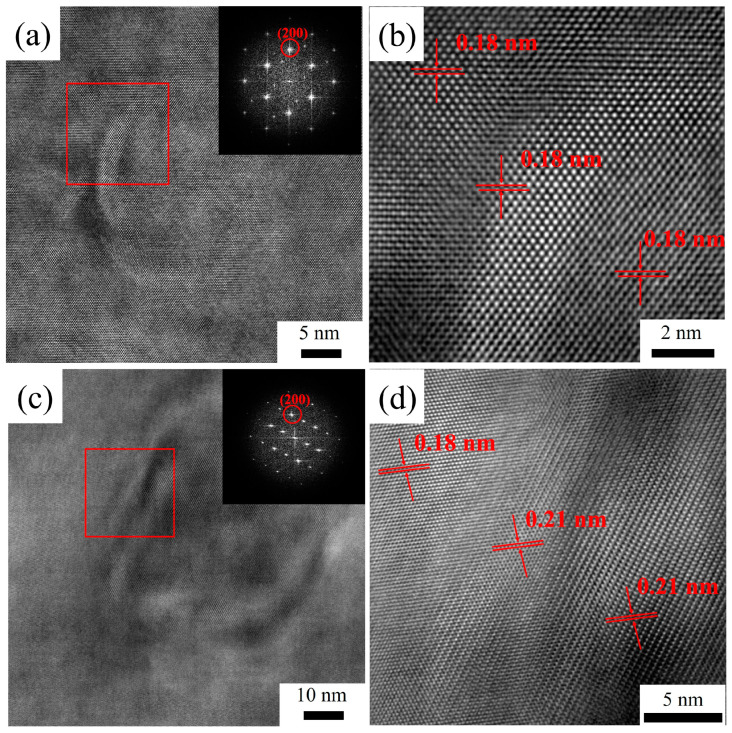
(**a**) HRTEM image after 400 °C annealing, (**b**) red frame region of HRTEM image after 400 °C annealing, (**c**) HRTEM image after 500 °C annealing, and (**d**) red frame region of HRTEM image after 500 °C annealing of AA5182-0.1Sc-0.1Zr.

**Figure 23 materials-16-05315-f023:**
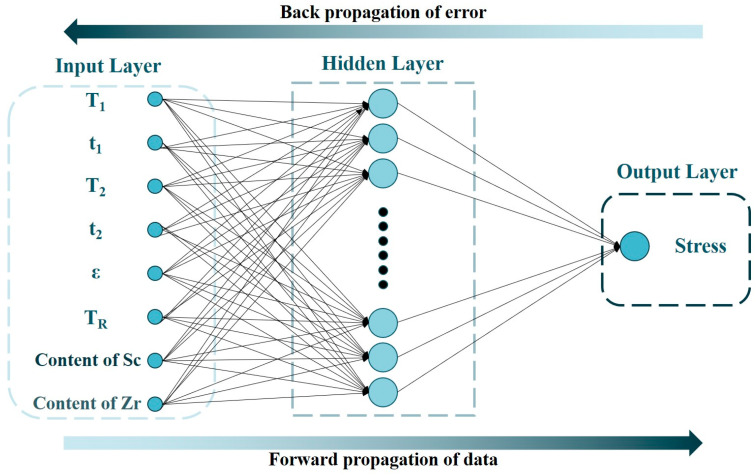
An artificial neural network model for predicting mechanical properties of 5182-Sc-Zr alloy at room temperature.

**Figure 24 materials-16-05315-f024:**
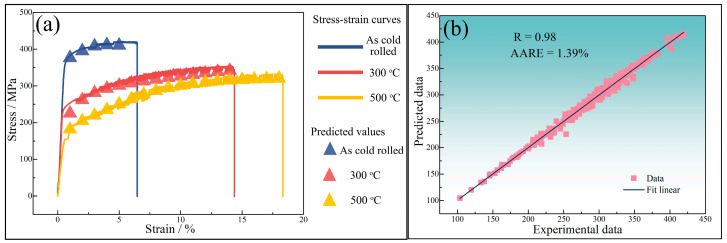
(**a**) ANN predicted value and stress–strain curve, (**b**) linear fitting relationship between predicted values and experimental values of 5182-0.1Sc-0.1Zr alloy.

**Figure 25 materials-16-05315-f025:**
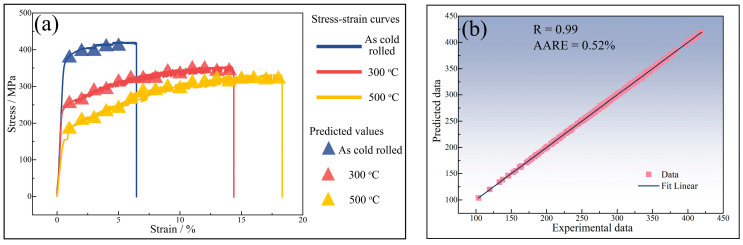
(**a**) GA-ANN predicted value and stress–strain curve, (**b**) linear fitting relationship between predicted values and experimental values of 5182-0.1Sc-0.1Zr alloy.

**Figure 26 materials-16-05315-f026:**
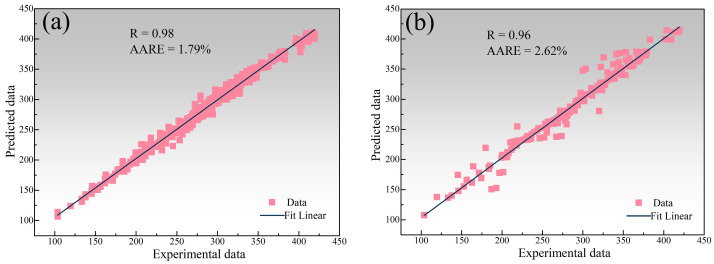
Comparison of predicted data and experimental data by cross-validation method in (**a**) training sets, (**b**) test sets.

**Table 1 materials-16-05315-t001:** Chemical composition of 5182-Sc-Zr alloys (wt.%).

Materials	Mg	Mn	Fe	Si	Sc	Zr	Al
AA5182	4.53	0.22	0.18	0.07	-	-	Bal.
AA5182-0.05Sc-0.1Zr	4.50	0.22	0.19	0.07	0.05	0.1	Bal.
AA5182-0.1Sc-0.1Zr	4.40	0.22	0.17	0.07	0.1	0.1	Bal.

**Table 2 materials-16-05315-t002:** Size of Al_3_(Sc_x_Zr_1−x_) dispersoids.

Annealing Temperature	250 °C	300 °C	400 °C	500 °C
dm (nm)	17.1	17.6	18.1	25

## Data Availability

No publicly archived dataset was created.

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
