# Peer review of "Determining Homogenization Parameters and Predicting 5182-Sc-Zr Alloy Properties by Artificial Neural Networks"

_materials, 2023, doi:10.3390/ma16155315_

Round 1

Reviewer 1 Report

In the paper "Determining homogenization parameters and predicting 5182-Sc-Zr alloy properties by Artificial neural networks", the authors construct the ANN-based model for the calculation of the microhardness and electroconductivity of the aluminum alloy after the homogenization. The constructed model has shown high accuracy with high Pearson coefficient values for all datasets. The paper also contains interesting microstructural findings. However, the paper is overloaded by an additional ANN-based model for tensile stress-strain curves and should be corrected accordingly following comments:

1.                 The part of references about ANN-based models in the Introduction part is too old. It is recommended to analyze new papers about modeling the alloys’ properties using the ANN approach (e.g., 10.3390/met13040664,  10.3390/met13040737, 10.3390/met12030447, etc).

2.                 Part 3.4 Establishment of ANN for room temperature performance of 5182-Sc-Zr alloy should be fully removed from the manuscript due to a large number of errors and the absence of the practical value. The scheme in Figure 23 is incorrect:

-                     The authors wrote that there are two hidden layers. However, only one hidden layer is present in the scheme. At the same time, the second hidden layer with just one neuron is useless.

-                     Accordingly, presented experimental data the input parameters were only the content of the Sc and Zr, the temperature of annealing after the deformation, and the strain during the testing. However, the authors present in the scheme other parameters.

3.                 The authors have just approximated 15 stress-strain curves by ANN-based model. Such a model has no scientific and practical value.

4.                 It is unclear, how the developed ANN-base model for the prediction of the microhardness and electroconductivity of the aluminum alloy after the homogenization may be used practically. Additional information should be added to the manuscript as values of the weights and biases and the form of the activation functions in neurons.

5.                 Minor correction:

-                     What software was used for the ANN-based model construction? The information should be added to the manuscript.

-                     The standard deviations should be added to the values of the electroconductivity in Figures 3 and 7.

-                     Line 389: “Berger vector” should be “Burgers vector”.

Author Response

Thank you for your letter and for the reviewers’ comments concerning our manuscript entitled “Determining homogenization parameters and predicting 5182-Sc-Zr alloy properties by Artificial neural networks” ID (Materials-2515052). Those comments are all valuable and very helpful for revising and improving our paper, as well as the important guiding significance to our researches. We have studied comments carefully and have made correction which we hope meet with approval. Revised portion are marked up using the “Track Changes” function in the manuscript. The comments and our corresponding responses were given below.

Reviewer #1:

Responds to the reviewer’s comments:

  1. Comment: The part of references about ANN-based models in the Introduction part is too old. It is recommended to analyze new papers about modeling the alloys’ properties using the ANN approach

Response:

Several papers from 2021 to 2023 using the ANN method were selected to replace the original literature 15 to 18.

“Hijazi used an artificial neural network in their study to predict the residual strength of an aluminum plate containing multiple site damage cracks, and the results demonstrated that a single neural network model can accurately predict the residual strength of all materials and structures[15]. Finocchiaro compared the performance of artificial neural network to multiple linear regression models in predicting the chemical stabil-ity of volcanic alkaline-activated materials, and found that the ANNs outperformed the multiple linear regression models in terms of predictive accuracy[16]. Dikici predicted the corrosion resistance and age-hardening behavior of gradient materials Al/TiC using ANN[17]. Ravi established an artificial neural network in their study to predict the hot deformation performance, and their findings demonstrated that it is possible to establish a high-accuracy ANN model even with a small dataset[18]” is rewritten as “In Jiang's study, the utilization of the backpropagation neural network in conjunction with the finite element method yielded outcomes for the accurate forecasting of 2A12 alloy grain size subsequent to extrusion[16]. In the research conducted by Kuppusamy on eco-friendly engineered geopolymer composites, the employment of an artificial neural network with cross-validation methodology aided in the formulation of the material design, which led to the realization of the intended compressive and tensile strength properties in the final product[17]. In the investigations pertaining to the thermal deformation properties of metals, researchers developed constitutive models, strain-compensated constitutive models, and artificial neural networks for the purpose of flow stress prediction. Through the assessment of the models' predictive capabilities using residual analysis, correlation coefficient (R), and average absolute relative error (AARE), it was determined that the artificial neural network exhibited superior accuracy compared to other models[18,19].”

  1. Jiang, H.S.; Wu, R.D.; Yuan, C.L.; Jiao, W.; Chen, L.L.; Zhou, X.Y. Prediction of Recrystallization Structure of 2A12 Aluminum Alloy Pipe Extrusion Process Based on BP Neural Network. Metals, 2023, 13(4), 664-683.
  2. Kuppusamy, Y.; Jayaseelan, R.; Pandulu, G.; Sathish Kumar, V.; Murali, G.; Dixit, S.; Vatin, N.I. Artificial Neural Net-work with a Cross-Validation Technique to Predict the Material Design of Eco-Friendly Engineered Geopolymer Compo-sites. Materials (Basel), 2022, 15(10), 3443-3463.
  3. Churyumov, A.; Kazakova, A.; Churyumova, T. Modelling of the Steel High-Temperature Deformation Behaviour Using Artificial Neural Network. Metals, 2022, 12(3), 447-459.
  4. Yang, H.; Bu, H.; Li, M.; Lu, X. Prediction of Flow Stress of Annealed 7075 Al Alloy in Hot Deformation Using Strain-Compensated Arrhenius and Neural Network Models. Materials (Basel), 2021, 14(20), 5986-5999.

  1. Comment: The scheme in Figure 23 is incorrect: The authors wrote that there are two hidden layers. However, only one hidden layer is present in the scheme. At the same time, the second hidden layer with just one neuron is useless.

Response: The model is reconstructed using a single hidden layer network.

Figure 23 is replaced with a new image, as shown below.

“The number of hidden layers in the network is 2, each containing 25 and 1 neuron, respectively.” is rewritten as “The network is a single hidden-layer network with 25 neurons.”

With the predicted data obtained from the new network, Figure 24 is modified, as shown below.

  1. Comment: Accordingly, presented experimental data the input parameters were only the content of the Sc and Zr, the temperature of annealing after the deformation, and the strain during the testing. However, the authors present in the scheme other parameters.

Response: When establishing the ANN used in this research, the input data include the temperature and time of the first stage homogenization, the temperature and time of the second stage homogenization, the content of Sc and Zr, the cold rolling strain and the annealing temperature, a total of eight elements. The figure below shows the number of elements in the dataset, which contains 264 groups of 8 elements each in the input dataset.

  1. Comment: The authors have just approximated 15 stress-strain curves by ANN-based model. Such a model has no scientific and practical value.

Response: When constructing the model, we obtained a dataset consisting of 264 sets of data by collecting data points through fixed strain intervals. This method refers to the relevant literature, some of which are listed below.

The purpose of this portion of the work is to investigate whether Artificial Neural Networks can be employed to develop models for predicting the influence of multithreaded machining or heat treatment on the final performance of a product. The research conducted by the research group encompasses various aspects of the 5182-Sc-Zr alloy, including its hot working performance, welding properties, corrosion resistance, and surface treatments. We aim to progressively establish a network for predicting the diverse properties of the alloy.

[1] H. Ahmadi, H.R. Rezaei Ashtiani, M. Heidari, A comparative study of phenomenological, physically-based and artificial neural network models to predict the Hot flow behavior of API 5CT-L80 steel, Materials Today Communications 25 (2020). “10.1016/j.mtcomm.2020.101528”

[2] S.A. Sani, G.R. Ebrahimi, H. Vafaeenezhad, A.R. Kiani-Rashid, Modeling of hot deformation behavior and prediction of flow stress in a magnesium alloy using constitutive equation and artificial neural network (ANN) model, Journal of Magnesium and Alloys 6(2) (2018) 134-144. “10.1016/j.jma.2018.05.002”

[3] H. Yang, H. Bu, M. Li, X. Lu, Prediction of Flow Stress of Annealed 7075 Al Alloy in Hot Deformation Using Strain-Compensated Arrhenius and Neural Network Models, Materials (Basel) 14(20) (2021) 5986-5999. “10.3390/ma14205986”

  1. Comment: It is unclear, how the developed ANN-base model for the prediction of the microhardness and electroconductivity of the aluminum alloy after the homogenization may be used practically. Additional information should be added to the manuscript as values of the weights and biases and the form of the activation functions in neurons.

Response: Based on the findings of this study, models for predicting the microhardness and electrical conductivity of the homogenized alloy can aid in the selection of homogenization parameters. When initially designing the homogenization experiment, we referred to the literature and selected 10 experimental temperatures within a large temperature range. However, during the experimental process, it was found that homogenization experiments were time-consuming due to experimental conditions and other factors. Therefore, we attempted to reduce the number of experiments while ensuring an adequate amount of data, and supplemented the missing experimental data with predicted data from an Artificial Neural Network. Ultimately, we found that the ANN was able to effectively capture the variations in alloy microhardness and electrical conductivity. Therefore, in future research, it is appropriate to use predicted data from an Artificial Neural Network (ANN) to assess the variations in material properties, while adhering to the principles of scientific methodology.

In the paper, I included the following words: " The network employed the Levenberg-Marquardt backpropagation algorithm."

The weights and other data among the neurons are represented in the matrix depicted in the following figure. I deemed the presentation of these data to be insignificant; hence, it was not included in the article.

  1. Comment: Minor correction

What software was used for the ANN-based model construction? The information should be added to the manuscript.

Response: The following content has been added to the manuscript. “The artificial neural networks established in this study were implemented with the assistance of MATLAB.”

The standard deviations should be added to the values of the electroconductivity in Figures 3 and 7.

Response: During the conductivity testing in this study, five different locations were selected to test each sample, with a maximum conductivity difference of less than or equal to 0.1 among the same sample. If there was a conductivity difference larger than 0.1 between the tested location and other locations of the same sample, several methods were employed to calibrate the data. These methods included recalibrating the conductivity meter, allowing the sample to rest for a period of time, and replacing the parallel sample. Due to the small errors and low variability in conductivity, standard deviations were not included in the figures.

Line 389: “Berger vector” should be “Burgers vector”.

Response: Errors on line 410 and line 415 have been fixed, “Berger vector” is rewritten as “Burgers vector”.

Reviewer 2 Report

- Line 191, line 275: Experimental data should be tabulated and compared with predicted data.

- Line477: Support the discussion with references

- Conclusion should be improved with more results, particularly ANN and GA models.

- References should be updated (2021-2023)

Author Response

Thank you for your letter and for the reviewers’ comments concerning our manuscript entitled “Determining homogenization parameters and predicting 5182-Sc-Zr alloy properties by Artificial neural networks” ID (Materials-2515052). Those comments are all valuable and very helpful for revising and improving our paper, as well as the important guiding significance to our researches. We have studied comments carefully and have made correction which we hope meet with approval. Revised portion are marked up using the “Track Changes” function in the manuscript. The comments and our corresponding responses were given below.

Reviewer #2:

  1. Comment: Line 191, line 275: Experimental data should be tabulated and compared with predicted data.

Response: Since the experimental data and predicted data are compared in Figure 4(b) and Figure 8(b), a comparison in a table is not employed again. Furthermore, due to the large number of hardness values (126) and conductivity values (72) in the two datasets, presenting them in a table format would occupy a significant amount of space, hence a table format is not utilized.

  1. Comment: Line477: Support the discussion with references.

Response: A reference on the Zener pinning effect has been added here.

[38] Wang, X.; Jiang, J.; Li, G.; Wang, X.; Shao, W.; Zhen, L. Particle-stimulated nucleation and recrystallization texture initiated by coarsened Al2CuLi phase in Al–Cu–Li alloy. Journal of Materials Research and Technology, 2021, 10, 643-650. “10.1016/j.jmrt.2020.12.046”

  1. Comment: Conclusion should be improved with more results, particularly ANN and GA models.

Response: The following content has been added in the conclusion:

“4. The predictive results of ANN for the mechanical properties of the 5182-Sc-Zr alloy were compared to the original experimental data, yielding a correlation coefficient of 0.98. The average absolute relative error was found to be 1.39%. Following the optimization by GA, the accuracy of the network was further improved, resulting in a reduction of the average absolute relative error between the predicted and experimental data to 0.52%.”

  1. Comment: References should be updated (2021-2023)

Response: Here are the references added, replacing some of the older references with more recent ones from 2021-2023:

  1. Jiang, H.S.; Wu, R.D.; Yuan, C.L.; Jiao, W.; Chen, L.L.; Zhou, X.Y. Prediction of Recrystallization Structure of 2A12 Aluminum Alloy Pipe Extrusion Process Based on BP Neural Network. Metals, 2023, 13(4), 664-683.
  2. Kuppusamy, Y.; Jayaseelan, R.; Pandulu, G.; Sathish Kumar, V.; Murali, G.; Dixit, S.; Vatin, N.I. Artificial Neural Net-work with a Cross-Validation Technique to Predict the Material Design of Eco-Friendly Engineered Geopolymer Compo-sites. Materials (Basel), 2022, 15(10), 3443-3463.
  3. Churyumov, A.; Kazakova, A.; Churyumova, T. Modelling of the Steel High-Temperature Deformation Behaviour Using Artificial Neural Network. Metals, 2022, 12(3), 447-459.
  4. Yang, H.; Bu, H.; Li, M.; Lu, X. Prediction of Flow Stress of Annealed 7075 Al Alloy in Hot Deformation Using Strain-Compensated Arrhenius and Neural Network Models. Materials (Basel), 2021, 14(20), 5986-5999.

Reviewer 3 Report

Dear authors;

The work entitled “Determining homogenization parameters and predicting 5182-Sc-Zr alloy properties by artificial neural networks” is very interesting. The manuscript is well written and well organized. The scientific part of this work is quite interesting.

I suggest that the authors should pay much attention to revise the manuscript by taking following points into account:

- The authors should more clearly emphasis the novelty of their work in the abstract and introduction.

- Abstract must include some explicit information about the results obtained rather than giving general statements.

- Line 11, page 1: “Artificial neural networks (ANNs)” instead of “ANNs”.

- Line 13, page 1: “ANNs” instead of “artificial neural networks”.

- Line 15, page 1: “electron backscatter diffraction (EBSD) and transmission electron microscopy (TEM)” instead of “EBSD and TEM”.

- Line 18, page 1: the authors should justify the choice of the homogenization parameters.

- Lines 47-49, page 2: the authors must provide the corresponding references.

- Line 57, page 2: “ANNs” instead of “artificial neural networks (ANNs)”.

- For the whole manuscript, the authors must leave a space between the value and the unit of measurement.

- Materials and Methods: concerning the measuring devices, the authors must provide their characteristics, the starting parameters, and the country of origin.

- L 127, page 4: “Zeiss focused ion beam scanning electron microscopes (FIB/SEM)” instead of “FIB/SEM”.

- Results and Discussion:

- The authors must add the error bars in all figures.

- A comparison with the existing published literature is missing.

- The resolution of all figures should be improved.

- Figure 11: It is necessary to add the corresponding element for each color.

- L 390, page 15: symbol should be written in Italic.

- Figure 21: you have to zoom in to clearly visualize the dislocations and dispersoids.

- L 508, page 20: “…alloy studied by Vo is less than 8 nm”. The sentence is not clear. Please correct it.

- L 516, page 20: Table 2. “Annealing temperature” instead of “size”.

Author Response

Thank you for your letter and for the reviewers’ comments concerning our manuscript entitled “Determining homogenization parameters and predicting 5182-Sc-Zr alloy properties by Artificial neural networks” ID (Materials-2515052). Those comments are all valuable and very helpful for revising and improving our paper, as well as the important guiding significance to our researches. We have studied comments carefully and have made correction which we hope meet with approval. Revised portion are marked up using the “Track Changes” function in the manuscript. The comments and our corresponding responses were given below.

Reviewer #3:

  1. Comment: The authors should more clearly emphasis the novelty of their work in the abstract and introduction.

Response: In order to emphasize the novelty, the following content is added to the Introduction:

“The majority of studies utilizing neural networks to establish models have predominantly focused on single production or experimental processes, thereby being limited to capturing the impact of processing conditions or heat treatment conditions on alloy performance within a singular process[16-19].”

  1. Comment: Abstract must include some explicit information about the results obtained rather than giving general statements.

Response:

“The artificial neural network was utilized for the selection of homogenization parameters (275 oC for 20h and 440 oC for 12h) for 5182-Sc-Zr alloy, and a highly accurate predictive model that includes multiple-threaded influencing factors was successfully established. The results confirm that Al3(ScxZr1-x) dispersoids strengthening in the alloy is through coherency strengthening and mod-ulus mismatch strengthening, which is determined by the coherent relationship between the Al3(ScxZr1-x) dispersoids and the Al matrix.” is rewritten as “The two-stage homogenization parameters were determined by studying the changes in microhardness and electrical conductivity of 5182-Sc-Zr alloy after homogenization with the assistance of artificial neural networks: the first-stage homogenization at 275 oC for 20 hours and the second-stage homogenization at 440 oC for 12 hours. The dispersoids have entirely precipitated after homogenization, and the alloy segregation has improved. A high-accuracy prediction model, incorporating multiple influencing factors through artificial neural networks, was successfully established to predict the mechanical properties of the 5182-Sc-Zr alloy after annealing. Based on the atomic plane spacing in HRTEM, it was determined that the Al3(ScxZr1-x) dispersoids and the Al matrix maintained a good coherence relationship after annealing at 400 oC. In this state, the strengthening effect of the Al3(ScxZr1-x) dispersoids is achieved through coherency strengthening and modulus mismatch strengthening.”

  1. Comment: Line 11, page 1: “Artificial neural networks (ANNs)” instead of “ANNs”.

Response: “ANNs” is rewritten as “Artificial neural networks (ANNs).

  1. Comment: Line 13, page 1: “ANNs” instead of “artificial neural networks”.

Response: This sentence has been deleted.

  1. Comment: Line 15, page 1: “electron backscatter diffraction (EBSD) and transmission electron microscopy (TEM)” instead of “EBSD and TEM”.

Response: “EBSD and TEM” is rewritten as “electron backscatter diffraction (EBSD) and transmission electron microscopy (TEM)”

  1. Comment: Line 18, page 1: the authors should justify the choice of the homogenization parameters.

Response: The following information has been added to the abstract of the manuscript:

“The two-stage homogenization parameters were determined by studying the changes in micro-hardness and electrical conductivity of 5182-Sc-Zr alloy after homogenization with the assistance of artificial neural networks: the first-stage homogenization at 275 oC for 20 hours, and the second-stage homogenization at 440 oC for 12 hours. The dispersoids have entirely precipitated after homogenization, and the alloy segregation has improved.”

  1. Comment: Lines 47-49, page 2: the authors must provide the corresponding references.

Response: The following literature is cited:

[10] Behler, J. Perspective: Machine learning potentials for atomistic simulations. APL Materials, 2016, 145(17), 170901.

  1. Comment: Line 57, page 2: “ANNs” instead of “artificial neural networks (ANNs)”.

Response: “Artificial neural networks (ANNs)” is rewritten as “ANNs”.

  1. Comment: For the whole manuscript, the authors must leave a space between the value and the unit of measurement.

Response: Space is added between values and units in the manuscript.

For example, “2mm” is rewritten as “2 mm”. “2h” is rewritten as “2 hours”.

  1. Comment: Materials and Methods: concerning the measuring devices, the authors must provide their characteristics, the starting parameters, and the country of origin.

Response: Information on measuring devices has been added.

“The electrical conductivity testing was performed by SIGMASCOPE SMP10.”

“The microhardness was obtained by Shimadzu HMV-G with a load of 0.2 kgf.”

  1. Comment: Line127, page 4: “Zeiss focused ion beam scanning electron microscopes (FIB/SEM)” instead of “FIB/SEM”.

Response: “FIB/SEM” is rewritten as “Zeiss focused ion beam scanning electron microscopes (FIB/SEM)”

  1. Comment: The authors must add the error bars in all figures.

Response: The experimental data for hardness already includes error bars, while the predicted data does not have error. Additionally, the reason for not including error bars in the conductivity data is as follows:

During the conductivity testing in this study, five different locations were selected to test each sample, with a maximum conductivity difference of less than or equal to 0.1 among the same sample. If there was a conductivity difference larger than 0.1 between the tested location and other locations of the same sample, several methods were employed to calibrate the data. These methods included recalibrating the conductivity meter, allowing the sample to rest for a period of time, and replacing the parallel sample. Due to the small errors and low variability in conductivity, error bars were not included in the figures.

  1. Comment: A comparison with the existing published literature is missing.

Response: The research in this paper refers to the previous research methods, but few studies involve the same field, that is, the establishment of ANN for the homogenization treatment and annealing treatment of 5182-Sc-Zr alloy, so there is no relevant comparison.

  1. Comment: The resolution of all figures should be improved.

Response: The "Do not compress images" function in Word is turned on.

  1. Comment: Figure 11: It is necessary to add the corresponding element for each color.

Response: After anodizing, aluminum alloys exhibit various colors when observed under a polarized microscope. When a uniform and well-distributed oxide film is formed on the anodized surface, rainbow colors may appear. This color phenomenon is known as interference colors, which result from the interference and reflection of light on the oxide film. Individual colors have no corresponding elements.

  1. Comment: L 390, page 15: symbol should be written in Italic.

Response: Because the formula in the journal template uses the Palatino Linotype, it has not been modified.

  1. Comment: Figure 21: you have to zoom in to clearly visualize the dislocations and dispersoids.

Response: Replace Figure 21 with new images.

  1. Comment: L 508, page 20: “…alloy studied by Vo is less than 8 nm”. The sentence is not clear. Please correct it.

Response:

“Vo has conducted a study to investigate the effect of precipitate diameter on the dis-persoid strengthening of Al-0.055Sc-0.005Er-0.02Z-0.05Si alloy. It was observed that as the precipitate diameter increased from 8 nm to 12 nm, the dispersoid strengthening decreased from 97 MPa to 62 MPa, following an inverse proportionality with the size of the dispersoid, in accordance with the Orowan strengthening law[40]. These results indicate that the critical size for the dispersoid strengthening mechanism transition in the Al-0.055Sc-0.005Er-0.02Z-0.05Si alloy studied by Vo is less than 8 nm.” is rewritten as “Vo conducted a study to investigate the effect of precipitate diameter on the dispersoid strengthening of the Al-0.055Sc-0.005Er-0.02Z-0.05Si alloy and observed that the dispersoid strengthening decreased from 97 MPa to 62 MPa as the precipitate diameter increased from 8 nm to 12 nm, following an inverse proportionality with the size of the dispersoid in accordance with the Orowan strengthening law[40]. These results suggest that the critical size for the transition of the dispersoid strengthening mechanism in the studied Al-0.055Sc-0.005Er-0.02Z-0.05Si alloy is less than 8 nm.”

  1. Comment: L 516, page 20: Table 2. “Annealing temperature” instead of “size”.

Response: “size” is rewritten as “Annealing temperature”

Round 2

Reviewer 1 Report

The authors have answered previous comments and improved the manuscript. The paper may be accepted for publication.

Reviewer 3 Report

Dear Authors;

Thank you for your work in revsising your manuscript according to the indicated comments and suggestions. The revised paper is well improved. 

Therefore, I hope this revised manuscript is now acceptable for publication.